# Data-Efficient Pipeline for Offline Reinforcement Learning with Limited Data

**Allen Nie**[*]    **Yannis Flet-Berliac**    **Deon R. Jordan**
**William Steenbergen**    **Emma Brunskill**
Department of Computer Science
Stanford University
*anie@stanford.edu

## Abstract

Offline reinforcement learning (RL) can be used to improve future performance by leveraging historical data. There exist many different algorithms for offline RL, and it is well recognized that these algorithms, and their hyperparameter settings, can lead to decision policies with substantially differing performance. This prompts the need for pipelines that allow practitioners to systematically perform algorithm-hyperparameter selection for their setting. Critically, in most real-world settings, this pipeline must only involve the use of historical data. Inspired by statistical model selection methods for supervised learning, we introduce a task- and method-agnostic pipeline for automatically training, comparing, selecting, and deploying the best policy when the provided dataset is limited in size. In particular, our work highlights the importance of performing multiple data splits to produce more reliable algorithm-hyperparameter selection. While this is a common approach in supervised learning, to our knowledge, this has not been discussed in detail in the offline RL setting. We show it can have substantial impacts when the dataset is small. Compared to alternate approaches, our proposed pipeline outputs higher-performing deployed policies from a broad range of offline policy learning algorithms and across various simulation domains in healthcare, education, and robotics. This work contributes toward the development of a general-purpose meta-algorithm for automatic algorithm-hyperparameter selection for offline RL.

## 1  Introduction

Offline/batch reinforcement learning has the potential to learn better decision policies from existing real-world datasets on sequences of decisions made and their outcomes. In many of these settings, tuning methods online is infeasible and deploying a new policy involves time, effort and potential negative impact. Many of the existing datasets for applications that may benefit from offline RL may be fairly small in comparison to supervised machine learning. For instance, the MIMIC intensive care unit dataset on sepsis that is often studied in offline RL has 14k patients (Komorowski et al., 2018), the number of students frequently interacting with an online course will often range from hundreds to tens of thousands (Bassen et al., 2020), and the number of demonstrations collected from a human operator manipulating a robotic arm is often on the order of a few hundred per task (Mandlekar et al., 2018). In these small data regimes, recent studies (Mandlekar et al., 2021; Levine et al., 2020) highlight that with limited data, the selection of hyperparameters using the training set is often challenging. Yet hyperparameter selection also has a substantial influence on the resulting policy's performance, particularly when the algorithm leverages deep neural networks.

One popular approach to address this is to learn policies from particular algorithm-hyperparameter pairs on a training set and then use offline policy selection, which selects the best policy given a validation set (Thomas et al., 2015a, 2019; Paine et al., 2020; Kumar et al., 2021). However, when

36th Conference on Neural Information Processing Systems (NeurIPS 2022).

| Common Practices | Non-Markov Env | Data Efficient (re-train) | Compare Across OPL | Considers Evaluation Variation | Considers Training Variation |
|---|---|---|---|---|---|
| **Policy selection (1 split)** | | | | | |
| Internal Objective / TD-Error (Thomas et al., 2015b, 2019) | (depends) | ✗ | ✗ | ✗ | ✗ |
| OPE methods (Komorowski et al. (2018); Paine et al. (2020) | (depends) | ✗ | ✓ | ✗ | ✗ |
| OPE + BCa Val. (Thomas et al., 2015b) | (depends) | ✗ | ✓ | ✓ | ✗ |
| BVFT (Xie and Jiang, 2021) | ✗ | ✗ | ✗ | ✗ | ✗ |
| BVFT + OPE (Zhang and Jiang, 2021) | ✗ | ✗ | ✓ | ✓ | ✗ |
| Q-Function Workflow (Kumar et al., 2021) | ✗ | ✓ | ✗ | ✗ | ✗ |
| **Ours: $\mathcal{A}_i$ selection (multi-split)** | | | | | |
| Cross-Validation | ✓ | ✓ | ✓ | ✓ | ✓ |
| Repeated Random Subsampling | ✓ | ✓ | ✓ | ✓ | ✓ |

Table 1: A summary of commonly used approaches for choosing a deployment policy from a fixed offline RL dataset. We define *Data Efficient* as: the approach assumes the algorithm can be re-trained on all data points; *(depends)* as: depends on whether the underlying OPL or OPE methods make explicit Markov assumption or not.

the dataset is limited in size, this approach can be limited: (a) if the validation set happens to have no or very few good/high-reward trajectories, then trained policies cannot be properly evaluated; (b) if the training set has no or very few such trajectories, then no good policy behavior can be learned through any policy learning algorithm; and (c) using one fixed training dataset is prone to overfitting the hyperparameters on this one dataset and different hyperparameters could be picked if the training set changes. One natural solution to this problem is to train on the entire dataset and compare policy performance on the same dataset, which is often referred to as the internal objective approach. In Appendix A.1 we conduct a short experiment using D4RL where this approach fails due to the common issue of Q-value over-estimation (Fujimoto et al., 2019).

There has been much recent interest in providing more robust methods for offline RL. Many rely on the workflow just discussed, where methods are trained on one dataset and Offline Policy Evaluation (OPE) is used to do policy selection (Su et al., 2020; Paine et al., 2020; Zhang and Jiang, 2021; Kumar et al., 2021; Lee et al., 2021; Tang and Wiens, 2021; Miyaguchi, 2022). Our work highlights the impact of a less studied issue: the challenge caused by data partitioning variance. We first motivate the need to account for train/validation partition randomness by showing the wide distribution of OPE scores the same policy can obtain on different subsets of data or the very different performing policies the same algorithm and hyperparameters can learn depending on different training set partitions. We also prove a single partition can have a notable failure rate in identifying the best algorithm-hyperparameter to learn the best policy.

We then introduce a general pipeline for algorithm-hyperparameters (AH) selection and policy deployment that: (a) uses repeated random sub-sampling (RRS) with replacement of the dataset to perform AH training, (b) uses OPE on the validation set, (c) computes aggregate statistics over the RRS splits to inform AH selection, and (d) allows to use the selected AH to retrain on the entire dataset to obtain the deployment policy. Though such repeated splitting is common in supervised learning, its impact and effect have been little studied in the offline RL framework. Perhaps surprisingly, we show that our simple pipeline leads to substantial performance improvements in a wide range of popular benchmark tasks, including D4RL (Fu et al., 2020) and Robomimic (Mandlekar et al., 2021).

## 2 Related work

**Offline Policy Learning (OPL).** In OPL, the goal is to use historical data from a fixed behavior policy $\pi_b$ to learn a reward-maximizing policy in an unknown environment (Markov Decision Process, defined in Section 3). Most work studying the sampling complexity and efficiency of offline RL (Xie and Jiang, 2021; Yin et al., 2021) do not depend on the structure of a particular problem, but empirical performance may vary with some pathological models that are not necessarily Markovian. Shi et al. (2020) have precisely developed a model selection procedure for testing the Markovian hypothesis and help explain different performance on different models and MDPs. To address this problem, it is inherently important to have a fully adaptive characterization in RL because it could save considerable time in designing domain-specific RL solutions (Zanette and Brunskill, 2019). As an answer to a variety of problems, OPL is rich with many different methods ranging from policy gradient (Liu et al., 2019), model-based (Yu et al., 2020; Kidambi et al., 2020), to model-free methods (Siegel et al., 2020; Fujimoto et al., 2019; Guo et al., 2020; Kumar et al., 2020) each based on different assumptions on the system dynamics. Practitioners thus dispose of an array of algorithms and corresponding hyperparameters with no clear consensus on a generally applicable evaluation tool for offline policy selection.

**Offline Policy Evaluation (OPE).** OPE is concerned with evaluating a target policy's performance using only pre-collected historical data generated by other (behavior) policies (Voloshin et al., 2021). Each of the many OPE estimators has its unique properties, and in this work, we primarily consider two main variants (Voloshin et al., 2021): Weighted Importance Sampling (WIS) (Precup, 2000) and Fitted Q-Evaluation (FQE) (Le et al., 2019). Both WIS and FQE are sensitive to the partitioning of the evaluation dataset. WIS is undefined on trajectories where the target policy does not overlap with the behavior policy and self-normalizes with respect to other trajectories in the dataset. FQE learns a Q-function using the evaluation dataset. This makes these estimators very different from mean-squared errors or accuracy in the supervised learning setting – the choice of partitioning will first affect the function approximation in the estimator and then cascade down to the scores they produce.

**Offline Policy Selection (OPS).** Typically, OPS is approached via OPE, which estimates the expected return of candidate policies. Zhang and Jiang (2021) address how to improve policy selection in the offline RL setting. The algorithm builds on the Batch Value-Function Tournament (BVFT) (Xie and Jiang, 2021) approach to estimating the best value function among a set of candidates using piece-wise linear value function approximations and selecting the policy with the smallest projected Bellman error in that space. Previous work on estimator selection for the design of OPE methods include Su et al. (2020); Miyaguchi (2022) while Kumar et al. (2021); Lee et al. (2021); Tang and Wiens (2021); Paine et al. (2020) focus on offline hyperparameter tuning. Kumar et al. (2021) give recommendations on when to stop training a model to avoid overfitting. The approach is exclusively designed for Q-learning methods with direct access to the internal Q-functions. On the contrary, our pipeline does policy training, selection, and deployment on any offline RL method, not reliant on the Markov assumption, and can select the best policy with potentially no access to the internal approximation functions (black box). We give a brief overview of some OPS approaches in Table 1.

## 3 Background and Problem Setting

We define a stochastic Decision Process $M = \langle \mathcal{S}, A, T, r, \gamma \rangle$, where $\mathcal{S}$ is a set of states; $A$ is a set of actions; $T$ is the transition dynamics (which might depend on the full history); $r$ is the reward function; and $\gamma \in (0, 1)$ is the discount factor. Let $\tau = \{s_i, a_i, s'_i, r_i\}_{i=0}^{L}$ be the trajectory sampled from $\pi$ on $M$. The optimal policy $\pi$ is the one that maximizes the expected discounted return $V(\pi) = \mathbb{E}_{\tau \sim \rho_\pi}[G(\tau)]$ where $G(\tau) = \sum_{t=0}^{\infty} \gamma^t r_t$ and $\rho_\pi$ is the distribution of $\tau$ under policy $\pi$. For simplicity, in this paper we assume policies are Markov $\pi : S \rightarrow A$, but it is straightforward to consider policies that are a function of the full history. In an offline RL problem, we take a dataset: $\mathcal{D} = \{\tau_i\}_{i=1}^{n}$, which can be collected by one or a group of policies which we refer to as the behavior policy $\pi_b$ on the decision process $M$. The goal in offline/batch RL is to learn a decision policy $\pi$ from a class of policies with the best expected performance $V^\pi$ for future use. Let $\mathcal{A}_i$ to denote an AH pair, i.e. an offline policy learning algorithm and its hyperparameters and model architecture. An offline policy estimator takes in a policy $\pi_e$ and a dataset $\mathcal{D}$, and returns an estimate of its performance: $\widehat{V} : \Pi \times \mathcal{D} \rightarrow \mathbb{R}$. In this work, we focus on two popular Offline Policy Evaluation (OPE) estimators: Importance Sampling (IS) (Precup, 2000) and Fitted Q-Evaluation (FQE) (Le et al., 2019) estimators. We refer the reader to Voloshin et al. (2021) for a more comprehensive discussion.

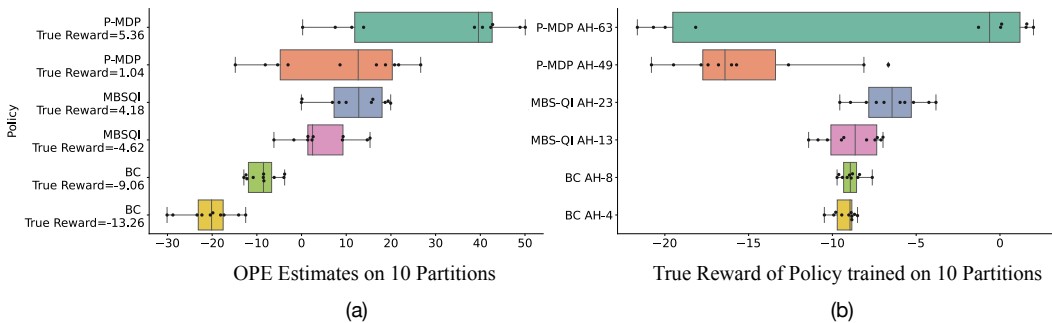

Figure 1: True performance and evaluation of 6 $\mathcal{A}_i$ pairs on the Sepsis-POMDP (N=1000) domain. (a) shows the OPE estimations and (b) shows the variation in terms of true performance. The variations are due to the different AH pairs of the policies *but also* to the sensitivity to the training/validation splits.

## 4 The Challenge of Offline RL $\mathcal{A}_i$ Selection

An interesting use-case of offline RL is when domain experts have access to an existing dataset (with potentially only a few hundred trajectories) about sequences of decisions made and respective outcomes, with the hope of leveraging the dataset to learn a better decision policy for future use. In this setting, the user may want to consider many options regarding the type of RL algorithm (model-based, model-free, or direct policy search), hyperparameter, or deep network architecture to use.

Automated algorithm selection is important because different $\mathcal{A}_i$ (different AH pairs) may learn very diverse policies, each with significantly different performance $V^{\mathcal{A}_i}$. Naturally, one can expect that various algorithms lead to diverse performance, but using a case-study experiment on a sepsis simulator (Oberst and Sontag, 2019), we observe in Figure 1(b) that the sensitivity to hyperparameter selection is also substantial (cf. different average values in box plots for each method). For example, MBS-QI (Liu et al., 2020) learns policies ranging from over -12 to -3 in their performance, depending on the hyperparameters chosen.

Precisely, to address hyperparameter tuning, past work often relies on executing the learned policies in the simulator/real environment. When this is not feasible, as in many real-world applications, including our sepsis dataset example, where the user may only be able to leverage existing historical data, we have no choice but to rely on off-policy evaluation. Prior work (Thomas et al., 2015b; Farajtabar et al., 2018; Thomas et al., 2019; Mandlekar et al., 2021) have suggested doing so using a hold-out method, after partitioning the dataset into training and validation sets.

Unfortunately, the partitioning of the dataset itself may result in substantial variability in the training process (Dietterich, 1998). We note that this problem is particularly prominent in offline RL where high-reward trajectories are sparse and affect both policy learning and policy evaluation. To explore this hypothesis, we consider the influence of the train/validation partition in the same sepsis domain, and we evaluate the trained policies using the Weighted Importance Sampling (WIS) (Precup, 2000) estimator. Figure 1(a) shows the policies have drastically different OPE estimations with sensitivity to randomness in the dataset partitioning. We can observe the same phenomena in Figure 1(b) with largely different true performances depending on the dataset splitting for most of the policies $\mathcal{A}_i$. This is also illustrated on the left sub-figure of Figure 4 where in the case where a single train-validation split is used, an $\mathcal{A}_i$ that yields lower-performing policies will often be selected over those that yield higher-performing policies when deployed.

### 4.1 Repeated Experiments for Robust Hyperparameter Evaluation in Offline RL

We now demonstrate why it is important to conduct repeated random sub-sampling on the dataset in offline RL. Consider a finite set of $J$ offline RL algorithms $\mathcal{A}$. Let the policy produced by algorithm $\mathcal{A}_j$ on training dataset $\mathcal{D}$ be $\pi_j$, its estimated performance on a validation set $\hat{V}^{\pi_j}$, and its true (unknown) value be $V^{\pi_j}$. Denote the true best resulting policy as $\pi_{j^*} = \arg\max_j V^{\pi_j}$ and the corresponding algorithm $\mathcal{A}_{j^*}$. Let the best policy picked based on its validation set performance as $\pi_{\hat{j}^*} = \arg\max_j \hat{V}^{\pi_j}$ and the corresponding algorithm $\mathcal{A}_{\hat{j}^*}$.

**Theorem 1.** *There exist stochastic decision processes and datasets such that (i) using a single train/validation split procedure that selects an algorithm-hyperparameter with the best performance*

*on the validation dataset will select a suboptimal policy and algorithm with significant finite probability, $P(\pi_{\hat{j}*} \neq \pi_{j*}) \geq C$, with corresponding substantial loss in performance $O(V_{max})$, and, in contrast, (ii) selecting the algorithm-hyperparameter with the best average validation performance across $N_s$ train/validation splits will select the optimal algorithm and policy with probability 1: $\lim_{N_s \to \infty} P(\pi_{\hat{j}*} = \pi_{j*}) \to 1$.*

*Proof Sketch.* Due to space constraints we defer the proof to Appendix A.3. Briefly, the proof proceeds by proof by example through constructing a chain-like stochastic decision process and considers a class of algorithms that optimize over differing horizons (see e.g. Jiang et al. (2015); Cheng et al. (2021); Mazoure et al. (2021)). The behavior policy is uniformly random meaning that trajectories with high rewards are sparse. This means there is a notable probability that in a single partition of the dataset, the resulting train and/or validation set may not contain a high reward trajectory, making it impossible to identify that a full horizon algorithm, and resulting policy, is optimal.

In the proof and our experiments, we focus on when the training and validation sets are of equal size. If we use an uneven split, such as $80/20\%$, the failure probability can further increase if only a single partition of the dataset is used. We provide an illustrative example in the Appendix. Note that Leave-one-out Cross-Validation (LooCV) will also fail in our setting if we employ, as we do in our algorithm, WIS, because as a biased estimator, WIS will return *the observed return of the behavior policy if averaging over a single trajectory, independent of the target policy to be evaluated*. We explain this further in Appendix A.11.

# 5 SSR: Repeated Random Sampling for $\mathcal{A}_i$ Selection and Deployment

In this paper, we are interested in the following problem: *If offline RL training and evaluation are very sensitive to the partitioning of the dataset, especially in small data regimes, how can we reliably produce a final policy that we are confident is better than others and can be reliably deployed in the real-world?*

Instead of considering the sensitivity to data partition as an inherent obstacle for offline policy selection, we view this as statistics to leverage for $\mathcal{A}_i$ selection. We propose a general pipeline: *Split Select Retrain* (SSR) (of which we provide a pseudo-code in Algorithm 1, Appendix A.4) to reliably optimize for a good deployed policy given only: an offline dataset, an input set of AH pairs and an off-policy evaluation (OPE) estimator. This deployment approach leverages the random variations created by dataset partitioning to select algorithms that perform better *on average* using a robust hyperparameter evaluation approach which we develop below.

First, we split and create different partitions of the input dataset. For each train/validation split, each algorithm-hyperparameter (AH) is trained on the training set and evaluated using the input OPE method to yield an estimated value on the validation set. These estimated evaluations are then averaged, and the best AH pair ($\mathcal{A}^*$) is selected as the one with the highest average score. Now the last step of the SSR pipeline is to re-use the entire dataset to train one policy $\pi^*$ using $\mathcal{A}^*$.

**Repeated Random Sub-sampling (RRS).** As Theorem 1 suggests, one should ensure a sufficient amount of trajectories in the evaluation partition to lower the failure rate $C$. We propose to create RRS train-validation partitions. This approach has many names in the statistical model selection literature, such as Predictive Sample Reuse Method (Geisser, 1975), Repeated Learning-Test Method (Burman, 1989) or Monte-Carlo Cross-Validation (Dubitzky et al., 2007). It has also been referred to as Repeated Data Splitting (Chernozhukov et al., 2018) in the heterogeneous treatment effect literature. We randomly select trajectories in $\mathcal{D}$ and put them into into two parts: $R^{\text{train}}$ and $R^{\text{valid}}$. We repeat this splitting process $K$ times to generate paired datasets: $(R_1^{\text{train}}, R_1^{\text{valid}}), (R_2^{\text{train}}, R_2^{\text{valid}}), ..., (R_K^{\text{train}}, R_K^{\text{valid}})$. We compute the generalization performance estimate as follows:

$$\mathcal{G}_{\mathcal{A}, \text{RS}_K} = \frac{1}{K} \sum_{k=1}^{K} \left[ \hat{V}(\mathcal{A}(R_k^{\text{train}}); R_k^{\text{valid}}) \right] \tag{1}$$

A key advantage of overlap partitioning is that it maintains the size of the validation dataset as $K$ increases. This might be favorable since OPE estimates are highly dependent on the state-action coverage of the validation dataset – the more data in the validation dataset, the better OPE estimators can evaluate a policy's performance. As $K \to \infty$, RRS approaches the leave-$p$-out cross-validation

(CV), where $p$ denotes the number of examples in the validation dataset. Since there are $\binom{n}{p}$ possible selections of $p$ data points out of $n$ in our dataset, it is infeasible to use exact leave-$p$-out CV when $p > 2$, but a finite $K$ can still offer many advantages. Indeed, Krzanowski and Hand (1997) point out that leave-$p$-out estimators will have lower variance compared to leave-one-out estimators, which is what the more commonly used M-fold cross-validation method converges to when $M = n - 1$. We discuss more in Appendix A.2.

## 6 Experiments

In this section, we answer the following questions: (a) how does the pipeline `SSR-RRS` compare to other methods? (b) does the proposed pipeline for $\mathcal{A}_i$ selection and policy deployment allow us to generate the best policy trained on the whole dataset? (c) does re-training on the whole dataset (data efficiency) generate better policies than policies trained on half of the dataset when $\mathcal{A}_i$ is selected by the pipeline? In addition, we conduct two ablation studies to answer to: what number of splits should we use for `SSR-RRS`, and what is the impact of dataset size on the pipeline results?

### 6.1 Task/Domains

The experimental evaluation involves a variety of real-world and simulated domains, ranging from tabular settings to continuous control robotics environments. We evaluate the performance of `SSR` in selecting the best algorithm regardless of task domains and assumptions on task structure. We conduct experiments on eight datasets (Figure 2) from five domains (details in Appendix A.14), which we give a short description below, and use as many as 540 candidate `AH` pairs for the Sepsis POMDP domain.

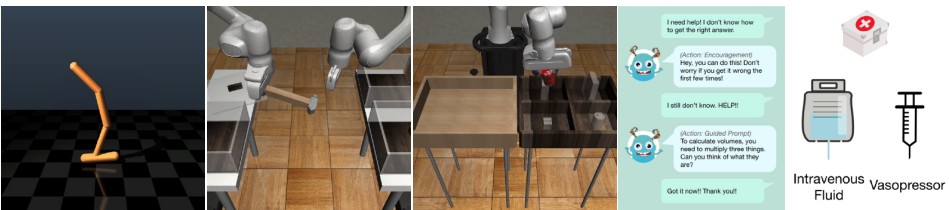

Figure 2: Illustrations from left to right of the D4RL, Robomimic, TutorBot and Sepsis domains.

**Sepsis.** The first domain is based on the simulator and work by Oberst and Sontag (2019) and revolves around treating sepsis patients. The goal of the policy for this simulator is to discharge patients from the hospital. In this domain, we experiment on two tasks: Sepsis-MDP and Sepsis-POMDP, a POMDP version of Sepsis-MDP.

**TutorBot.** The second domain includes a TutorBot simulator that is designed to support 3-5th grade elementary school children in understanding the concept of volume and engaging them while doing so. An online study was conducted using a policy-gradient-based RL agent, which interacted with about 200 students. We took the observations from this online study and built a simulator that reflects student learning progression, combined with some domain knowledge.

**Robomimic.** Robomimic (Mandlekar et al., 2021) is composed of various continuous control robotics environments with suboptimal human data. We use the Can-Paired and Transport dataset composed of 200 mixed-quality human demonstrations. Mandlekar et al. (2021) attempted to use the RL objective loss on a 20% split validation set to select the best `AH` pair, but reported that the selected `AH` did not perform well in the simulator, which makes this task an interesting testbed for our pipeline.

**D4RL.** D4RL (Fu et al., 2020) is an offline RL standardized benchmark designed and commonly used to evaluate the progress of offline RL algorithms. We use 3 datasets (200k samples each) with different qualities from the Hopper task: hopper-random from a randomly initialized policy, hopper-medium from a policy trained to approximately 1/3 the performance of a policy trained to completion with SAC ("expert"), and hopper-medium-expert from a 50-50 split of medium and expert data.

### 6.2 Baselines

**One-Split OPE.** The simplest method to train and verify an algorithm's performance without access to any simulator is to split the data into a train $\mathcal{D}_{\text{train}}$ and valid set $\mathcal{D}_{\text{valid}}$. All policies are trained

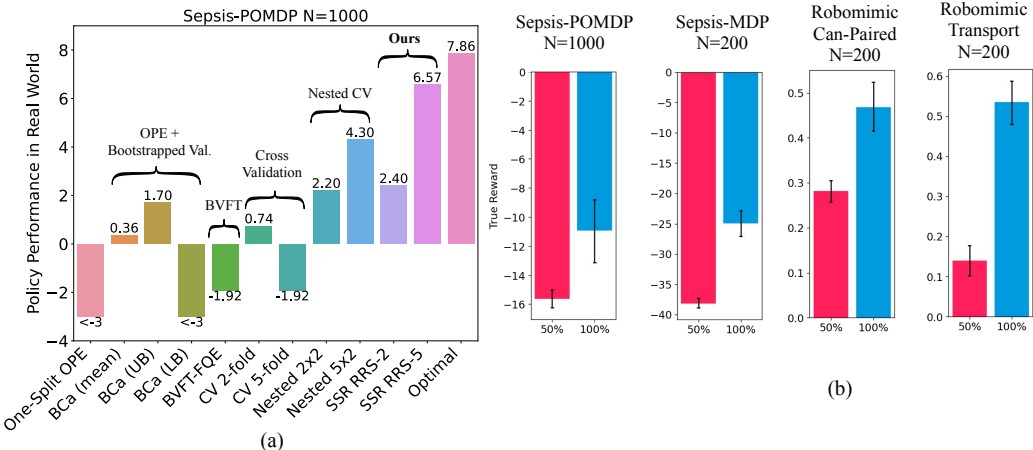

Figure 3: **(a)** We first compare our proposed pipeline to various other policy selection approaches in the Sepsis POMDP task. Our approach `SSR-RRS` 5-split consistently obtains policies that on average perform close to the optimal policy, significantly outperforming other approaches. **(b)** We investigate the importance of re-training in the small (N=200) to medium (N=1000) data regime. We show the true reward obtained by all policies from all `AH` pairs either trained on 50% of the data or 100% of the data. 95% confidence intervals are depicted as error bars. Most policies achieve higher rewards when trained on more data, even more so when the dataset is small or when tasks are more difficult (Robomimic).

on the same training set and evaluated on the same valid set. As we explained before, this method has the high potential of overfitting the chosen hyperparameter for one data partition – it might pick the best policy, but does not guarantee we can use the same hyperparameter to re-train on the full dataset.

**OPE on Bootstrapped Val.** Bootstrapping is a popular re-sampling technique that estimates prediction error in supervised learning models (Efron, 1983, 1986). The idea of using bootstrapping for OPE estimate is first utilized in HCOPE (Thomas et al., 2015b). Compared to the one-split method and the `SSR` pipeline, bootstrapping trains all policies on the same training dataset, and only considers variations in the validation set by creating bootstrapped samples. We refer to the considered Bias-corrected accelerated (BCa) bootstrap method as **BCa** in the experiments.

**Cross-Validation (CV).** One other natural alternative of repeated experiment validation is the popular M-fold Cross-Validation method (Stone, 1974). M-fold CV constructs a non-overlapping set of trajectories from the original dataset. For example, a 5-fold CV will train a policy on 80% of data and evaluate the policy on 20% of data, as it divides the dataset into 5 non-overlapping partitions. However, as we increase the number of splits $M$, which allows us to train/test our algorithms under more data split variations, each non-overlapping set $\mathcal{D}_m$ becomes smaller. When $M = n - 1$, M-fold CV becomes leave-one-out CV (LooCV). In this extreme case, many OPE estimators will not work properly, as we have shown in Appendix A.3. We further investigate a variant of M-fold CV called Nested M-fold CV (**Nested CV**), which repeats the M-fold non-overlapping partitioning K times. This procedure is computationally very expensive. Considering the fairness of comparison and computational efficiency, we only evaluate $K \times 2$-fold CV.

**OPE with BVFT.** Batch Value Function Tournament is the closest competitor to our method, which is a meta-algorithm for hyperparameter-free policy selection (Xie and Jiang, 2021; Zhang and Jiang, 2021). For a set of Q-value functions, BVFT makes pairwise comparisons of each (tournament-style) to select the best out of the entire set based on the BVFT-Loss. Compared to our method, BVFT incurs $\mathcal{O}(J^2)$ comparison given $J$ `AH` pairs, practically infeasible for large $J$. The original BVFT can only compare Q-functions, therefore only usable with OPL that directly learns Q-functions. Zhang and Jiang (2021) offers an extension to BVFT by using BVFT to compare between FQEs, therefore allowing BVFT to be OPL-agnostic. We adopt the two strategies recommended by the paper. Given $J$ `AH` pairs and $B$ FQEs, strategy 1 compares $J \times B$ FQE's Q-functions jointly ($\pi$ **x FQE**) and strategy 2 compares $B$ FQEs within each `AH` and pick the best FQE as the estimate of the `AH`'s value estimate ($\pi$ **+ FQE**). We discuss more in Appendix A.6 (calculations) and A.7 (time complexity).

### 6.3 Training and Evaluation

**Offline Policy Learning.** In the following, we outline a variety of Offline RL algorithms used in the evaluation of the `SSR` pipeline to demonstrate the generality of our approach and that it can reliably

| Category | Algorithms | Internal Q-function |
|---|---|---|
| Imitation Learning | BC (Pomerleau, 1991)
BCRNN (Mandlekar et al., 2018) | ✗ |
| Conservative Model-Free | BCQ (Fujimoto et al., 2019), CQL (Kumar et al., 2020),
IRIS (Mandlekar et al., 2020) | ✓ |
| Policy Gradient | POIS (Metelli et al., 2018), BC+POIS (ours), BC+mini-POIS (ours) | ✗ |
| Offline Model-Based | MOPO (Yu et al., 2020), P-MDP (ours) | ✓ |

Table 2: List of AH we are comparing in our experiments.

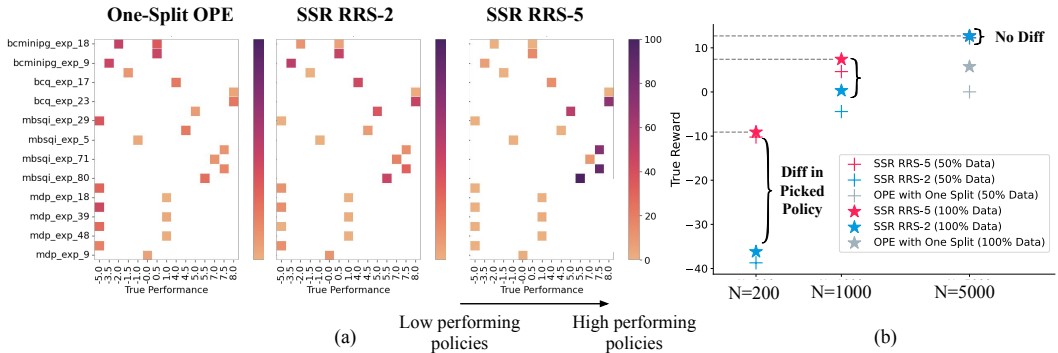

(a)

(b)

Figure 4: **(a)** We show the effect of choosing $K$ (the number of train/valid splits) for SSR-RRS. We ablate $K$ and run the simulation 500 times. The heatmaps shows the frequency at which each policy is chosen by our method and its final performance in the environment. In this experiment, SSR-RRS chooses over 540 AH pairs. As we can see, when the number of splits $K$ is larger, SSR-RRS consistently pick better policies. **(b)** In the Sepsis-POMDP domain, we show that as the number of trajectories ($N$) in the offline dataset increases, data partitioning becomes less important. Though RRS still outperforms the 1-split policy selection.

produce the optimal final policy using the selected AH pair. This marks a departure from workflows designed for specific algorithms, such as Kumar et al. (2021). We experiment with popular offline RL methods (see Table 2 and we provide algorithmic and hyperparameter details in Table A.2). **Offline Policy Evaluation.** We use WIS estimators for the tabular and discrete action domains: Sepsis and TutorBot. We use FQE for continuous action domains: Robomimic and D4RL. For each task, with a given dataset, we use the splitting procedure described in Section 5 to generate the partitioning. We describe how we compute the results for each figure in Appendix A.13.

## 7 Results

$\mathcal{A}_i$ **Selection Comparison.** In Figure 3(a), we compare five approaches (One-Split OPE, BCa, BVFT-FQE, CV, and SSR-RRS) on the Sepsis-POMDP domain with 1000 patients. For each approach, we compute a score per AH pair, and select the best algorithm according to each. For fairness in comparison, all selected $\mathcal{A}_i$ are re-trained on the full dataset and we report the final performance in the real environment. As expected, **One-Split OPE** performed the worst. Surprisingly, using the lower bound of bootstrapped (**BCa (LB)**) confidence interval also does not allow to pick good policies, LB being perhaps too conservative. We see that **CV 2-fold** and **CV 5-fold** do not perform well either. **CV 2-fold** does not allow enough repetition and **CV 5-fold** makes the validation set size too small. We observe clearly that SSR-RRS **5-split** performs the best and selected policies that are on average very close to the optimal policy's performance. **BVFT-FQE** relies on FQE, which is a misspecified model on the Sepsis domain and difficult to optimize given the small dataset size; hence it does not select good policies in Sepsis. However, in Robomimic Can-Paired and D4RL Hopper, BVFT-FQE is able to pick good policies, albeit not significantly better or worse than other methods, and still worse than SSR-RRS **5-split** in the mixed (more realistic) "medium-expert" dataset. We show more analysis of BVFT compared to our method in the Appendix. Table 3 aggregates the results for all the considered domains in our study. Our approach SSR-RRS **5-split** is distinctly able to more consistently select policies that, once deployed, perform close to the optimal policy across all tasks.

| Re-trained on full dataset | BVFT-FQE $\pi$ x FQE | BVFT-FQE $\pi$ + FQE | CV-2 | CV-5 | SSR RRS-2 | SSR RRS-5 | Optimal Policy |
|---|---|---|---|---|---|---|---|
| **Robomimic:** | | | | | | | |
| Can-Paired | 0.65 | 0.71 | 0.72 | 0.72 | 0.71 | **0.73** | 0.75 |
| Transport | 0.21 | 0.0 | 0.42 | 0.42 | 0.62 | **0.70** | 0.74 |
| **D4RL (Hopper):** | | | | | | | |
| random | 321.75 | 317.72 | **325.37** | **325.37** | 324.92 | **325.37** | 325.37 |
| medium | 934.71 | 1227.81 | 1227.81 | 1304.53 | 1296.87 | **1304.54** | 1392.93 |
| medium-expert | 2677.93 | 2677.93 | 2530.04 | 2530.04 | 3481.34 | **3657.80** | 3657.80 |
| **Sepsis:** | | | | | | | |
| MDP (n=200) | — | -19.32 | -10.26 | -20.32 | -13.01 | **-7.85** | -1.94 |
| POMDP (n=1000) | — | -1.92 | 0.74 | -1.92 | 2.40 | **6.75** | 7.86 |
| **TutorBot:** | | | | | | | |
| POMDP (n=200) | — | — | 1.34 | 1.19 | 1.30 | **1.38** | 1.43 |

Table 3: Comparison of the performance obtained by a policy deployed using the SSR pipeline vs. using 1-split policy selection approaches on a wide range of application domains. Cells = average true return. We note that ($\pi$ x FQE) is very computationally expensive when we search through a large AH space (in Sepsis and TutorBot), therefore we exclude them.

**The Benefits of Re-training Policies Selected with SSR.** In Figure 3(b), we plot the true reward of a selected policy $\mathcal{A}_i$ when only trained on 50% of the dataset (the training set) compared to when trained on 100% of the dataset. As expected, in the small data regime, every single trajectory matters. Policies trained on the full dataset significantly outperform policies trained only on half of it. This experiment provides strong evidence in favor of AH selection (done with RRS on the full dataset) over policy selection (done on the training set) in offline RL.

**The Impact of Number of Repeats for SSR-RRS.** The proposed pipeline SSR-RRS has a hyperparameter $K$ for the number of repeated data splitting. In Figure 4(a), we show the true performance of the policy that is being selected by SSR-RRS with $K = 1, 2, 5$ by running 500 simulations with heatmaps on the frequency of each policy is selected. We observe that when $K = 1$ (equivalent to the One-Split OPE method), policies are picked quite uniformly; many of which are performing poorly. When $K = 5$, higher-performing policies are selected much more frequently. From Table 3, we conclude that $K = 5$ generally works well across various domains. Naturally, the number of split $K$ will be chosen in line with the computing budget available; $K = 5$ appears to be a reasonable choice.

**The Impact of Dataset Size.** Finally, we investigate to which extent the proposed pipeline is necessary when the dataset size is sufficiently large. We use the Sepsis-POMDP domain with 200, 1000, and 5000 patients. We show the best policies that are most frequently selected by our approach in Figure 4(b). Unsurprisingly, policies trained on larger datasets perform better. In the 200-patient dataset, having SSR-RRS **5-split** is crucial in picking the best policy, as most policies perform quite poorly. The gap between different approaches becomes smaller with 1000 patients, and even smaller when there are 5000 patients in the dataset. However, it is worth noting that even in the large dataset regime (N=5000), SSR-RRS still outperforms the One-Split OPE method in selecting the best algorithm.

**Additional Analysis.** Our method SSR-RRS can also be used to select hyperparameters for a single algorithm, as we demonstrate in Appendix A.9. One might also wonder how sensitive is SSR-RRS pipeline to the choice of OPE method used inside the pipeline. OPE methods are known to significantly vary in accuracy for different domains, and unsurprisingly, using a reasonable OPE method for the domain is important (see Appendix A.8). Note that the OPE estimators we use in our results are very popular ones, and it is possible to use standard approaches, though additional benefits may come from using even better OPE methods. Finally, related to this question, one might wonder if particular OPE methods might be biased towards certain OPL algorithms which make similar assumptions (such as assuming a Markov structure): interestingly in preliminary experiments, FQE estimators did not seem to give FQI algorithms higher performance estimations (see Appendix A.10).

## 8 Discussion and Conclusion

We presented SSR, a pipeline for training, comparing, selecting and deploying offline RL policies in a small data regime. The approach performs automated AH selection with a robust hyperparameter

evaluation process using repeated random sub-sampling. SSR allows to consistently and reliably deploy best-performing policies thanks to jointly avoiding overfitting on a single dataset split and being data efficient in re-using the whole dataset for final training. We prove that a single split has a high failure rate of discovering the optimal AH because of reward sparsity. We have demonstrated its strong empirical performance across multiple and various challenging domains, including real-world applications where AH tuning cannot be performed online.

There exist many interesting areas for future work. The proposed offline RL pipeline assumes the user/practitioner has selected a particular OPE method. OPE is an important subarea of its own and different approaches have different bias/variance tradeoffs. Recent work on automated model selection algorithms for OPE (Su et al., 2020; Lee et al., 2021) are a promising approach for producing good internal estimators. A second issue is that while our approach aims to produce a high-performing policy, it does not also produce an accurate estimate of this policy since the entire dataset is used at the end for training. An interesting issue is whether cross-splitting (Chernozhukov et al., 2016) or other methods could be used to compute reliable estimators as well as perform policy optimization.

## 9   Acknowledgment

Research reported in this paper was supported in part by a Hoffman-Yee grant, NSF grant #2112926 and the DEVCOM Army Research Laboratory under Cooperative Agreement W911NF-17-2-0196 (ARL IoBT CRA). The views and conclusions contained in this document are those of the authors and should not be interpreted as representing the official policies, either expressed or implied, of the Army Research Laboratory or the U.S.Government. The U.S. Government is authorized to reproduce and distribute reprints for Government purposes notwithstanding any copyright notation herein. We would like to thank Jonathan N. Lee, Henry Zhu, Matthew Jorke, Tong Mu, Scott Fleming, and Eric Zelikman for discussions.

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
