# A Appendix

## A.1 Prelude Experiment

In this section, we put ourselves in a situation where model selection would be performed by comparing different `AH` pairs on their internal objective or value function estimates on a given dataset, as described near the beginning of Section 1. We use three datasets of different qualities (random, medium, and medium-expert) of the popular Hopper task from the D4RL benchmark (see Appendix A.14 for a detailed description) to train a total of 36 policies with different `AH` pairs and then calculate the resulting TD-Errors and Q-values on the whole dataset at the end of training.

To evaluate the performance one would obtain by employing such an approach to select the best policy, we report in Table A.1 the performance (true return in the environment) of the selected policies and compare them with the performance of the optimal policy for each of the datasets. The policies are selected either by finding the one which corresponds to the lowest TD-Error, or the one which corresponds to the highest Q-value. We also include the Kendall rank correlation coefficient (Gilpin, 1993) for each of the ranking methods (ranking with respect to TD-Error or Q-value) compared with the "true ranking" of policies ranked with respect to the performance in the environment:

$$\tau = \frac{(\text{ number of concordant pairs }) - (\text{ number of discordant pairs })}{\binom{n}{2}}$$

where n is the number of policies, and where "concordant pairs" are pairs from the two compared rankings for which the sort order agrees. A coefficient of 1 means the agreement between the two rankings is perfect.

| | TD-Error | | Q-value | | |
| --- | --- | --- | --- | --- | --- |
| | **Policy Selected (True Return)** | Kendall | **Policy Selected (True Return)** | Kendall | **Optimal Policy (True Return)** |
| **random** | 334.24 | -0.09 | 333.65 | -0.15 | 345.39 |
| **medium** | 1475.82 | 0.42 | 2381.37 | 0.21 | 2469.81 |
| **medium-expert** | 327.97 | -0.18 | 327.97 | -0.09 | 3657.80 |

Table A.1: Average Return (True Return obtained in the simulator) of the policy selected with respect to min(TD-Error) or max(Q-value) on the training dataset with a comparison to the True Return obtained by the Optimal Policy. Kendall rank correlation coefficient when ranking with respect to the same metrics. Policies are **trained and validated on the same dataset**. Task: Hopper.

Unsurprisingly, Table A.1 shows that one cannot rely on this straightforward pipeline to select a best-performing `AH` pair. Actually, for most of the datasets (the medium-expert dataset should resemble the most to what a dataset would look like in a real-world situation as it is composed of both high-quality and medium-quality data), following such an approach would produce and deploy a very bad performing policy.

## A.2 Connection between Leave-p-Out CV and RRS

Our RSS is a finite approximation of Leave-p-out (Lp0) cross-validation[1]. LpO is known in supervised learning, but rarely used due to the computational burden. The correctness of LpO is proved itecelisse2014optimal in a supervised learning setting with projection estimators. Unlike K-fold cross-validation, Leave-p-out CV selects $p$ data points for evaluation and the rest for training. In our proposed RSS method, we set $p = n/2$, and instead of exhaustively enumerating all possible selections of $p$ data points out of n data points, we only repeat this process $K$ times. Asymptotically as the amount of data goes to infinity, this approach should be correct, but also a single train/test split will also be correct in such a setting. The key challenges arise in the finite data setting, where the choice of dataset partitioning is key.

---

[1] https://scikit-learn.org/stable/modules/generated/sklearn.model_selection.LeavePOut.html

### A.3 Proof of Theorem 1

Consider a finite set of $J$ offline RL algorithms $\mathcal{A}$. Let the policy produced by algorithm $\mathcal{A}_j$ on training dataset $\mathcal{D}$ be $\pi_j$, its estimated performance on a validation set $\hat{V}^{\pi_j}$, and its true (unknown) value be $V^{\pi_j}$. Denote the true best resulting policy as $\pi_{j^*} = \arg\max_j V^{\pi_j}$ and the corresponding algorithm $\mathcal{A}_{j^*}$. Let the best policy picked based on its validation set performance as $\pi_{\hat{j}^*} = \arg\max_j \hat{V}^{\pi_j}$ and the corresponding algorithm $\mathcal{A}_{\hat{j}^*}$.

**Theorem 1.** *Then there exist stochastic decision processes and datasets such that (i) using a single train/validation split procedure will select a suboptimal policy and algorithm with significant finite probability, $P(\pi_{\hat{j}^*} \neq \pi_{j^*}) \geq C$, with corresponding substantial loss in performance $O(V_{max})$, and, in contrast, (ii) averaging across $N_s$ train/validation splits will select the optimal policy with probability 1: $\lim_{N_s \to \infty} P(\pi_{\hat{j}^*} = \pi_{j^*}) \to 1$.*

*Proof.* We proceed by constructing a stochastic decision process. A common domain to illustrate the importance of strategic exploration is a chain MDP. Here consider an episodic, finite horizon, finite chain, deterministic decision process with 6 states, $s_1, \ldots, s_H$, ($H = 6$) with two actions. $a_1$ moves the state one down except for at the starting state, and $a_2$ increments the state one up except for the final state: more formally, $p(s_{i-1}|s_i, a_1) = 1$ except for $p(s_1|s_1, a_1) = 1$; $p(s_{i+1}|s_i, a_2)$ except for $p(s_H|s_H, a_2) = 1$. The reward is 0 in all states except $R(s_1) = 1/6$ and $R(s_H) = 201$. All episodes are length $H = 6$ and start in state $s_1$. The optimal policy always takes $a_2$ and achieves $V_{max} = R(s_H)$. Any other policy achieves at most $H * 1/6 = 1$ reward.

The behavior policy is uniform random over the two actions, $\pi_b(a_1|s) = 0.5 = \pi_b(a_2)$. Let the available offline dataset $D$ consist of 200 episodes gathered using $\pi_b$. Given the behavior policy, each of the $64 = 2^H$ unique trajectories has an equal probability of being observed, and only one of these $\tau_h = (s_1, 0, a_2, s_2, 0, a_2, , s_3, 0, a_2, s_4, 0, a_2, s_5, a_2, s_H, R(s_H))$ achieves the highest return. On average out of 200 episodes[2], $n_{\tau_h} = 3 (= round(|\mathcal{D}|/(2^H)))$ episodes will match $\tau_h$. All other episodes will have a return of 1 or less.

Let there be a set of $H$ offline RL algorithms $\mathcal{A}_h$, each which optimizes the reward over a different horizon $h = 1 : H$, by constructing a maximum-likelihood estimate (MLE) MDP model $\mathcal{M}$ given a training dataset $D_{tr}$, and then computing a policy $\pi_h$ that optimizes the $h$-step value given the learned MDP $\mathcal{M}$ model[3] For example, algorithm $\mathcal{A}_2$ will take the MLE MDP model and construct a policy to optimize the sum over rewards for the next two time steps $\pi_2(s) = \arg\max_a r(s) + \sum_{s'} p(s'|s, a) r(s')$. We think this is a reasonable set of algorithms to consider as an illustrative example: the horizon length can directly influence the amount of data needed to compute an optimal policy, and recent work has explored using shorter horizons (Cheng et al., 2021; Mazoure et al., 2021; Liao et al., 2020), so choosing the right horizon can be viewed as a bias/variance tradeoff, suitable for automatic model selection.

Observe that even if given access to the true (unknown) MDP parameters, algorithms $\mathcal{A}_1, \ldots, \mathcal{A}_{H-1}$ will compute a policy that is suboptimal: due to the shortened horizon length, to optimize the expected total reward, the resulting policy computed for $s_1$ will be $\pi_h(s_1) = \pi_1(s_1) = a_1$ for these algorithms $\mathcal{A}_h$, $h = 1 : H - 1$. As the MDP is deterministic, this will also be true for any input dataset.

We now consider the impacts of partitioning the input dataset into a training dataset $D_{tr}$ taken as input by each algorithm $\mathcal{A}_h$ to compute a policy $\pi_{\hat{h}}$, and an evaluation/test dataset $D_{te}$: $D = D_{tr} \cup D_{te}$. For algorithm $\mathcal{A}_H$ to learn the optimal policy $\pi^*$ which achieves $V_{max}$, it must learn over a dataset $D_{tr}$ that includes one or more examples of the highest return trajectory $\tau_h$. Note that a single episode of $\tau_h$ in the training set is sufficient to learn the optimal policy[4].

---

[2]Our calculations can easily be extended to cases where there are different numbers of observed $\tau_h$, but for simplicity we assume a dataset where the average expected number of $\tau_h$ are observed.

[3]During planning with the learned MDP model, we restrict taking the maximum value over actions for a given state $s$ to only actions that have been taken at least once in that state in the dataset, e.g. $\max_{a \, s.t. \, n(s,a) \geq 1}$, where $n(s, a)$ is the counts of the number of times action $a$ was taken in state $s$ in the dataset. Note that in a finite dataset, some states and/or actions may not be observed, and this common choice simply ensures that the algorithm does not overestimate the value of untried actions.

[4]A single example of $\tau_h$ will induce a MLE $\hat{\mathcal{M}}$ with the correct reward model for all states, and the dynamics model for action $a_2$. From the procedure used to compute an optimal policy $\hat{\mathcal{M}}$, this will result in an optimal policy.

Assume that the offline evaluation of the policies learned by the algorithm on $D_{tr}$ is performed using importance sampling on $D_{te}$: note, our results will still apply, with minor modifications, if off policy evaluation is performed on $D_{te}$ using fitted Q evaluation (Le et al., 2019) or using a certainty-equivalent MDP constructed from $D_{te}$.

Then the off policy evaluation of the policy learned by the full horizon algorithm $\mathcal{A}_H$, $\hat{V}^{\pi_H}(s_1)$, will only be greater than 1 if there also exists at least one episode of the highest return trajectory $\tau_h$.

Assume the training dataset and validation dataset are constructed by randomly sampling 50% of the episodes to be in each. By assumption, there are $n_{\tau_h}$ samples of $\tau_h$, which have an equal chance of being in either the training or validation set. There are $n_{\tau_h} + 1$ ways to partition the $n_{\tau_h}$ exchangable episodes of $\tau_h$ into the training and validation sets, here $([3,0],[2,1],[1,2],[0,3])$. Note the training and validation set are identical in size ($|\mathcal{D}|/2$ trajectories each), and we only care about whether a trajectory $\tau$ is identical to $\tau_h$ or not. The probability that each of these partitions occurs is : $P([3,0]) = P([0,3]) = \frac{100}{200} * \frac{99}{199} * \frac{98}{198} \approx 0.123$.

From the above analysis, $\mathcal{A}_H$ can only learn an optimal policy, and its estimated value $\hat{V}^{\pi_5} > 1$ on $D_{te}$ if there is at least one $\tau_h$ in both the training and validation set datasets, which occurs in partitions $([2,1],[1,2])$. This occurs with probability 0.754. Otherwise, either (a) $\mathcal{A}_H$ will not learn an optimal policy, and instead will learn $\pi_H(s_1) = \pi_1(s_1) = a_1$, or (b) $\mathcal{A}_H$ will learn an optimal policy $\pi_H(s_1) = a_2$ but as the validation dataset does not contain $\tau_h$, $\hat{V}^{\pi_H} = 1/H < \hat{V}^{\pi_1}$. In both cases, the selected policy given its performance on the validation set will be $\pi_1(s_1) = a_1$. The resulting loss in performance is $V_{max} - V^{\pi_1} = V_{\max} - 1 = O(V_{\max})$. This failure occurs with substantial probability 24.6%. This proves part (i) of the theorem.

To prove part (ii) of the proof, we consider cases where at least one $\tau_h$ is in both $D_{tr}$ and $D_{te}$. Note $\hat{V}^{\pi_1} \leq \frac{R(s_1)}{1/2^H} = \frac{1}{1/2^H}$. Define $E_{ss}$ as a "successful split": the event that 1 or more of $\tau_h$ (high returns) episodes are in $D_{te}$, but not all $n_{\tau_h}$. On event $E_{ss}$, the optimal policy (which will be computed by $\mathcal{A}_H$ on the training set), will have an estimated value on $D_{te}$, using importance sampling:

$$\hat{V}_{ss}^{\pi^*} \geq \frac{1}{|D_{te}|} \frac{R(s_H)}{1/2^H} = \frac{1}{1/2^H} * \frac{201}{100} > 2\hat{V}^{\pi_1} \tag{2}$$

since there are at least 1 $\tau_h$ trajectories, each with propensity weight $\frac{1}{1/2^H}$ and reward $R(s_H)$. Therefore on Event $E_{ss}$ the optimal policy can be learned and estimated as having high reward. The probability of event $E_{ss}$ is greater than 0.5: $P(E_{ss}) = 0.754$.

In the repeated train-validation split setting, the algorithm selected is the one that has the best performance on the validation set, on average across all $N_s$ splits. Let $E_h$ be the event that at least half the train-validation dataset splits are successful (Event $E_{ss}$ holds for that split). In this case then the average performance of $\mathcal{A}_5$ will be at least

$$
\begin{aligned}
\hat{V}_{\mathcal{A}_5} &\geq \frac{1}{N_s}\left(\frac{N_s}{2}\hat{V}_{ss}^{\pi^*} + 0\right) \\
&\geq \frac{1}{N_s}\left(\frac{N_s}{2}2\hat{V}^{\pi_1} + 0\right) \\
&= \hat{V}^{\pi_1},
\end{aligned}
$$

where the first line uses a lower bound of 0 when the event $E_{ss}$ fails to hold, and substitutes in Equation 2. Therefore as long as event $E_h$ holds, the optimal policy $\pi^*$ (which will be computed by algorithm $\mathcal{A}_H$ will be selected. Since $P(E_{ss}) > 0.5$, the probability[5] as the number of splits goes to infinity that $E_{ss}$ holds on least half of those splits goes to 1: $\lim_{N_s \to \infty} P(E_h) \to 1$.

$\square$

---

[5]**calculate for finite S.

## A.4 SSR pseudo-code

---

**Algorithm 1** SSR-RRS: $\mathcal{A}_i$ Selection with Repeated Random Sub-sampling

---

**Input:** offline RL data $\mathcal{D}$; set of AH pairs $[\mathcal{A}_1, \mathcal{A}_2, ..., \mathcal{A}_z]$, OPE estimator $\widehat{V}$, split number $K \in \mathbb{N}$.

**Output:** policy $\hat{\pi}^*$ for deployment

$\mathcal{R} = \emptyset$
**for** $i \leftarrow 1...K$ **do**
    $R_i^{\text{train}}, R_i^{\text{valid}} = \texttt{Subsample}\,(\mathcal{D}, 0.5)$
    $\mathcal{R} = \mathcal{R} \cup (R_i^{\text{train}}, R_i^{\text{valid}})$
**end**
$\mathcal{G} = []$
**for** $i \leftarrow 1...z$ **do**
    $\mathcal{S} = []$
    **for** $j \leftarrow 1...K$ **do**
        $\pi_i = \mathcal{A}(R_j^{\text{train}})$
        $\mathcal{S}_{ij} = \widehat{V}(\pi_i; R_j^{\text{valid}})$
    **end**
    $\mathcal{G}_i = \frac{1}{K} \sum_{j=1}^{K} \mathcal{S}_{ij}$
**end**
$\mathcal{A}^* = \mathcal{A}_{o+}$ where $o = \arg\max(\mathcal{G})$
$\pi^* = \mathcal{A}^*(\mathcal{D})$
**return** $\pi^*$

---

## A.5 Code

We include the implementation and experiment code here: https://github.com/StanfordAI4HI/Split-select-retrain

## A.6 Experiment Detail Summary

We choose different sets of algorithms to evaluate our pipeline in every domain to demonstrate the generality of our approach and because some algorithms have limitations inherent to certain types of domains to which they can be applied. We list them in Table A.2.

Running a large number of algorithm-hyperparameter pairs many times is very computationally expensive. In order to save time and resources, we leverage the fact that multiple approaches can share resources. We describe how we compute the numbers for each approach as follows:

For each offline RL dataset in Sepsis, TutorBot, Robomimic, and D4RL, we produce the following partitions (we refer to this as the "partition generation procedure"):

1. 2-fold CV split (2 partitions consisted of $(S_i)$)
2. 5-fold CV split (5 partitions consisted of $(S_i)$)
3. 5 RRS split (5 partitions consisted of $(R_i^{\text{train}}, R_i^{\text{valid}})$)

Here, we briefly describe how to use these data partitions to select algorithms with alternative approaches.

**One-Split OPE.** The One-Split OPE method can be conducted to train and evaluate an algorithm on any of the RRS splits being produced, but only look at one split, without considering other splits. We let for a particular $i$, we let $\mathcal{D}_{\text{train}} = R^{\text{train}_i}$ and $\mathcal{D}_{\text{valid}} = R_i^{\text{valid}}$.

**BCa Bootstrap.** Similar to the One-Split OPE method, we can use RRS split for bootstrap. For a particular $i$, we let $\mathcal{D}_{\text{train}} = R^{\text{train}_i}$ and $\mathcal{D}_{\text{valid}} = R_i^{\text{valid}}$. Bootstrapping will re-sample with

| Experiment Domain | Number of Trajectories (N) | Average Trajectory Length | Number of Transitions in Total | AH Pairs Evaluated | Algorithms in Experiment |
|---|---|---|---|---|---|
| Sepsis-POMDP | 200 | 14 | 2792 | 540 | BC, POIS, BC+POIS, BC+mini-POIS, BCQ, MBSQI, pMDP, MOPO |
| Sepsis-POMDP | 1000 | 14 | 13708 | 540 | BC, POIS, BC+POIS, BC+mini-POIS, BCQ, MBSQI, pMDP, MOPO |
| Sepsis-POMDP | 5000 | 14 | 68576 | 148 | BC, POIS, BCQ, MBS-QI, pMDP, MOPO |
| Sepsis-MDP | 200 | 14 | 2792 | 383 | BC, BCQ, MBSQI, pMDP, POIS, BC + POIS BC+mini-POIS |
| TutorBot | 200 | 5 | 987 | 81 | BC, POIS, BC+POIS, BC+mini-POIS |
| Robomimic Can-Paired | 200 | 235 | 47,000 | 35 | BC, BCRNN, CQL, IRIS, BCQ |
| Robomimic Transport | 200 | 470 | 94,000 | 10 | BC, BCRNN, CQL, IRIS, BCQ |
| D4RL Hopper | 500 | 1000 | 500,000 | 4 x 4 | BCQ |
| D4RL HalfCheetah | 500 | 1000 | 500,000 | 4 x 4 | BCQ |

Table A.2: List of algorithms being used in which domain. 4 x 4 means we evaluate 4 AH pairs for the policy learning and 4 AH pairs for the policy evaluation estimators (FQE).

replacement on trajectories in $\mathcal{D}_{\text{valid}}$ to create (largely) overlapping subsets $B_1, B_2, ..., B_N$, with $|B_i| = n$. We then evaluate $\pi_e$ on each subset using $\widehat{V}$. The final score is computed through a bias correction process with an added acceleration factor (BCa).

**Nested K $\times$ 2-fold Cross-Validation.** We can also use the RRS split partitions to produce $K \times 2$ Nested CV by taking one RRS split $(R_i^{\text{train}}, R_i^{\text{valid}})$ by doing the following procedure:

$$s_i = \frac{\widehat{V}(\mathcal{A}(R_i^{\text{train}}); R_i^{\text{valid}}) + \widehat{V}(\mathcal{A}(R_i^{\text{valid}}); R_i^{\text{train}})}{2} \tag{3}$$

$$\mathcal{G}_{\mathcal{A}, \text{NCV}_K} = \frac{1}{K} \sum_{i=1}^{K} s_i \tag{4}$$

Intuitively, for $K \times 2$ Nested CV, we just need to swap the train and valid set produced by repeated sub-sampling and average to produce the algorithm performance score for a particular split $i$. Then we average the scores to get a final score for the algorithm.

**2-fold Cross-Validation.** Similar to the $K \times 2$ Nested CV, we can choose the $i$-th partition generated by the 10 RRS split procedure, and compute the score according to Equation 3. We do this for the Sepsis and TutorBot domains, but we do not do this for the Robomimic domain.

**Batch Value Function Tournament (BVFT)** Xie and Jiang (2021); Zhang and Jiang (2021) proposed to use pairwise Q-function comparisons to select the optimal Q-function from a set of Q-functions. Given $Q_i, Q_j$, let $\mathcal{G}_{ij}$ be the piecewise constant function class induced by binning $(s, a)$ and $(s', a')$ if $Q_i(s, a) = Q_j(s', a')$. Given an offline dataset $D$, we can compute the BVFT loss as

follow:

$$\hat{\mathcal{T}}_{\mathcal{G}_{ij}} Q := \underset{g \in \mathcal{G}_{ij}}{\arg\min} \frac{1}{|D|} \sum [(g(s,a) - r - \gamma \max_{a'} Q(s',a'))^2] \tag{5}$$

$$\mathcal{E}_{\epsilon_k}(Q_i, Q_j) = \|Q_i - \hat{\mathcal{T}}_{\mathcal{G}_{ij}} Q_j\|_{2,D} \tag{6}$$

$$\mathcal{E}_{\epsilon_k}(Q_i) = \max_j \mathcal{E}_{\epsilon_k}(Q_i, Q_j) \tag{7}$$

Zhang and Jiang (2021) proposed a method to automatically search through different discretization resolutions ($\epsilon_k$). In our experiment, we search through $[0.1, 0.2, 0.5, 0.7, 1.0, 3.0, 10.0]$. We use the BVFT code provided by Xie and Jiang (2021). Because BVFT can only compare Q-functions, Zhang and Jiang (2021) offered two strategies to perform policy selection for any model/algorithm. Here we briefly describe two strategies:

- Strategy 1 ($\pi$ x FQE): if we have 4 policies, and each policy is evaluated by 4 FQEs, then this strategy will compare 16 Q-functions (4 $\pi$ x 4 FQE).

- Strategy 2 ($\pi$ + FQE): if we have 4 policies, and each policy is evaluated by 4 FQEs, then this strategy will first run BVFT to compare 4 Q-functions (1 $\pi$ x 4 FQE), select the best Q-function for each $\pi$ (4 $\pi$ x 1 FQE), then we select the best policy by the average Q-value computed by each FQE.

We generally find strategy 2 more computationally efficient (because it makes a smaller number of comparisons). BVFT generally has $O(J^2)$ time complexity where $J$ is the number of Q-functions that need to be compared – it's easy to see that $16^2 = 256$ is much larger than $4^2 = 16$.

Our repeated experiment protocol (RRS) is reliant on choosing a good FQE. In order to compare fairly, for $\pi$ x FQE strategy, we only use the optimal FQE (the ones used in RRS and CV and one-split). We can see that in this condition, BVFT can do pretty well (even outperforming RRS in the D4RL-Hopper medium setting). For $\pi$ + FQE, because it focuses on the selection of FQE, we try 4 different FQE hyperparameters. We discuss this more in D4RL Experiment Details (in Section A.19).

## A.7 Computational Complexity

Most of the approaches we discussed in Section A.6 leverage multiple repetitions (resampling) to account for data allocation randomness. We provide a time complexity table below and define the following terms:

- H = number of AH pairs to evaluate
- N = total data samples. We assume the training time for each trajectory is $N_1$ and evaluation time for each trajectory is $N_2$, where $N = N_1 + N_2$
- M = number of folds in multi-fold cross-validation
- B = number of bootstraps (this number is 100 in our experiment)
- P = number of resolutions for BVFT's grid (proposed in Zhang and Jiang (2021))
- F = number of FQE hyperparameters (proposed in Zhang and Jiang (2021))

For BVFT, one can amortize the computational cost by caching (storing $Q(s,a)$ for all $(s,a)$ in the dataset). If caching is done only once, we treat the actual computation time for the validation data set as $n_2$. P is usually between 5 and 10. When H is relatively large, for example, H = 540 (in our experiment), H * H = 2.916e5. It's easy to see that RRS is slightly more expensive than M-Fold CV but less expensive than the pairwise comparison tournament algorithm (BVFT). Zhang and Jiang (2021) proposed BVFT-FQE that only makes pairwise tournament comparison between FQE hyperparameters – F is 5 in our experiments. It's also worth noting that BCa has a high evaluation cost when B is large – when B = 100, BCa evaluation cost is significantly higher than CV and RRS.

## A.8 Sensitivity to OPE Methods

OPE is often a critical part of OPL, which has motivated significant research into OPE. Thus the employed OPE method will likely impact the performance of our proposed pipeline. As has been

|  | Training Complexity | Evaluation Complexity |
|---|---|---|
| One-Split | $H \times N_1$ | $H \times N_2$ |
| Bootstrapping (BCa) | $H \times N_1$ | $H \times B \times N_2$ |
| M-Fold Cross-Validation | $(H \times M \times N \times (M\text{-}1))/M$ $= H \times N \times (M\text{-}1)$ | $(H \times M \times N \times 1)/M$ $= H \times N$ |
| K-Repeat RRS | $H \times K \times N_1$ | $H \times K \times N_2$ |
| BVFT (Xie and Jiang, 2021) | $H \times N_1$ | $(H \times H) \times N_2$ or $(H \times H) \times n_2$ |
| BVFT-auto (Zhang and Jiang, 2021) | $H \times N_1$ | $P \times (H \times H) \times N_2$ or $P \times (H \times H) \times n_2$ |
| BVFT-FQE (Zhang and Jiang, 2021) | $H \times N_1$ | $P \times H \times (F \times F) \times N_2$ or $P \times H \times (F \times F) \times n_2$ |

demonstrated in a recent bake-off paper (Voloshin et al., 2021), minimal-assumption OPE methods like weighted doubly robust methods (e.g. Jiang et al. (2015); Thomas and Brunskill (2016)) may be most consistently accurate for many domains. However if the domain is known to be Markov and the models are well specified, FQE methods will likely be more accurate in small data regimes.

To explore further the impact of the choice of OPE method, we conducted an additional experiment on the Sepsis-POMDP domain. The aim to was to look at the sensitivity of `SSR-RRS` for picking the best `AH` to the choice of OPE estimators. In addition to the prior OPE methods used in the main text, we included clipped IS (importance sampling), CWPDIS (Thomas and Brunskill, 2016), and 8 different FQE OPE variants, in which different networks, learning rate and epochs were used.

| Sepsis-POMDP | Parameters | Best AH Performance Chosen by SSR-RRS K=5 |
|---|---|---|
| FQE-1 | [64], lr=3e-4, epoch=20 | 2.84 |
| FQE-2 | [64], lr=1e-5, epoch=20 | -74.26 |
| FQE-3 | [64], lr=3e-4, epoch=50 | -20.88 |
| FQE-4 | [64], lr=1e-5, epoch=50 | -14.16 |
| FQE-5 | [128], lr=3e-4, epoch=20 | -75.26 |
| FQE-6 | [128], lr=1e-5, epoch=20 | -14.48 |
| FQE-7 | [128], lr=3e-4, epoch=50 | -75.54 |
| FQE-8 | [128], lr=1e-5, epoch=50 | -74.26 |
| IS | N/A | 4.47 |
| CWPDIS | N/A | 4.68 |
| WIS | N/A | 6.75 |

Table A.3: Using different OPE estimators in the `SSR-RRS` pipeline. FQE-1 denotes the FQE with the optimal FQE hyperparameter (heuristically chosen).

First, using FQE does generally much worse in this setting which is not very surprizing: FQE assumes the domain is Markov, which Sepsis-POMDP is not.

All importance-sampling based OPE methods yield quite similar performing algorithm-hyperparameter choices in this setting.

While there are some clear differences, if some basic information about the domain is known (Markov or not), it is likely possible to select a pretty good OPE. In addition, prior work has proposed heuristics (Voloshin et al., 2021) or automatic methods for automatic OPE selection (Su et al., 2020; Lee et al., 2021). An interesting direction for future work would be to include such methods in the pipeline.

We highlight that while it is well known that OPE methods are important, our paper focused on an under-explored issue: that the dataset partitioning can also introduce a substantial amount of *additional* impact on learning good policies / selecting good AH.

### A.9 Robustness of SSR-RRS

In Table 3, we only show the performance of the best policy among all AH pairs. Here we show that SSR-RRS can still robustly select a good hyperparameter for a given offline RL policy learning algorithm (the gap between best AH selected and true best AH is relatively small).

| Sepsis-POMDP | Range of True Policy Performance (95%CI) | Percentile of AH Chosen by SSR-RRS | Performance of AH Chosen by SSR-RRS | True Best AH Performance |
|---|---|---|---|---|
| BCQ | [-10.8, -0.73] | 94% | 5.98 | 7.86 |
| MBSQI | [-7.34, -2.26] | 95% | 6.40 | 7.42 |
| BC | [-8.98, -8.37] | 58% | -8.46 | -7.42 |
| BC+PG | [-5.55, -4.26] | 78% | -3.68 | 2.52 |
| P-MDP | [-31.17, -21.26] | 83% | 0.23 | 2.82 |

Table A.4: We show the relative position (percentile) of the AH selected by SSR-RRS K=5 pipeline.

For each algorithm, we evaluate over 24 to 72 hyperparameters, and we compute the 95% confidence interval of all these policies' true performance. Except for behavior cloning, we are picking hyperparameters that are out-performing 78%-95% of other hyperparameters in the same algorithm.

### A.10 Is FQE biased towards FQI algorithms?

In our evaluation on the Sepsis domain, FQE is used to evaluate both BCQ and MBSQI (both FQI-based) and BC and BCPG (policy-gradient algorithms).

We designed the following analysis experiment using our logged results. We first rank all AH pairs (540 of them) with their true performance in the simulator, and then we count the percentage of FQI (BCQ, MBSQI) algorithms that appear in the top 10%, 20%, and 50% percentile. The number in each cell should be read as: "90.7% of AH pairs in the top-10% based on True Performance are FQI-based". If FQE is biased towards FQI algorithms, we expect to see a higher percentage of BCQ and MBSQI AH pairs selected than the true performance baseline and compared to other OPE methods.

| Sepsis-POMDP OPE Method | % of BCQ and MBSQI AHs in Top-10% AHs | % of BCQ and MBSQI AHs in Top-20% AHs |
|---|---|---|
| True Performance | 90.7% | 61.1% |
| FQE-1 | 0% | 0% |
| WIS | 9.4% | 35.5% |
| RRS-5 WIS | 68.5% | 58.3% |

Table A.5: Examining whether FQE as an estimator will prefer FQI policy learning algorithms.

Based on this analysis, we believe that FQE is not biased to select FQI-based algorithms in the Sepsis-POMDP domain. However, our analysis is limited to one domain and only on two FQI-based algorithms. Further investigation is needed but beyond the scope of our paper.

### A.11 Additional Discussions

**Sensitivity to K in small and large datasets** In general, we expect the issue of data partitioning into a train and test split is most important in small datasets: as the dataset gets very large, a single train/test split will generally work well. Therefore, we suggest using a larger K for smaller datasets, but for larger datasets, a smaller K will likely be sufficient. Using our theoretical example in the

appendix (chain-MDP), this can also be observed – with a larger N, the failure probability for smaller numbers of repeats decreases. This N-K tradeoff has computational benefits if there is a limited computational budget (larger datasets will require more training, therefore, harder to use a larger K).

**Weighted importance sampling (WIS) as a biased estimator**     WIS is a self-normalizing importance sampling estimator. We refer readers to Owen (2013) Chapter 9 for a more detailed discussion on the statistical properties of this type of estimator. In Section 4.1 (line 174), we state:

> WIS will return the observed return of the behavior policy if averaging over a single trajectory, independent of the target policy to be evaluated.

In brief, WIS works by first computing the probability of the dataset trajectory appearing under the evaluation policy and behavior policy:

$$w_i = \prod_{t=1}^{L} \frac{\pi_e(a_t|s_t)}{\pi_b(a_t|s_t)}$$

Then, this coefficient is normalized before multiplying with the trajectory return, therefore:

$$\text{WIS}(D) = \frac{1}{n} \sum_{i=1}^{n} \frac{w_i}{\sum_{j=1}^{n} w_j} (\sum_{t=1}^{L} \gamma^t R_t^i).$$

Perhaps surprisingly, if there is a single trajectory, $n = 1$, this implies

$$\text{WIS}(D) = \frac{w_i}{w_i} (\sum_{t=1}^{L} \gamma^t R_t^i) = \sum_{t=1}^{L} \gamma^t R_t^i.$$

Here WIS is a biased estimator that returns the trajectory weighted reward, independent of $w_i$.

## A.12   Additional Experiment

We report the D4RL HalfCheetah result over the same setting as D4RL Hopper, where the result is averaged over 20 runs.

| Re-trained on full dataset | BVFT $\pi$ x FQE | BVFT $\pi$ + FQE | CV-2 | CV-5 | SSR RRS-2 | SSR RRS-5 | Optimal Policy |
|---|---|---|---|---|---|---|---|
| **D4RL (HalfCheetah):** | | | | | | | |
| random | -1.14 | 1106.94 | -1.13 | -1.13 | **1922.07** | **1922.07** | 1922.07 |
| medium | **4421.95** | 4290.33 | 4290.33 | 4290.33 | 4290.33 | 4290.33 | 4517.96 |
| medium-expert | 8118.84 | 8799.66 | 8118.84 | 8118.84 | **9681.78** | **9681.78** | 10364.36 |

Table A.6: Additional comparison of the performance obtained by a policy deployed using the SSR pipeline vs. using 1-split policy selection approaches on D4RL HalfCheetah. Cells = average true return.

## A.13   Figure Generation Procedure

Given our partition generation procedure, there are some methods (One-Split OPE, $K \times 2$ Nested CV, and SSR-RRS $K$ when $K < 5$) that have a few different partitions to choose from. For example, out of the 5 RRS split partitions, which partition should we choose for the One-Split OPE method? If we choose one partition, and the One-Split method cannot select the best algorithm, does that mean the One-Split method is bad, or could the 9 other partitions do better for the One-Split method? In order to evaluate these approaches fairly, we exhaustively train and evaluate on the 5 RRS splits, swap the train/valid set, and train/evaluate on them again, generating 20 scores. For the aforementioned methods, we randomly sample from these 10 (or 20, if Nested CV is being evaluated) scores to simulate the setting that we happen to get one particular split. We run this sampling procedure multiple times and compute the average performance of the policies that are chosen by conditioning on one or $K$ particular partitions.

### A.14 Domain Descriptions

**Sepsis.** The first domain is based on the simulator and works by Oberst and Sontag (2019) and revolves around treating sepsis patients. The goal of the policy for this simulator is to discharge patients from the hospital. There are three treatments the policy can choose from antibiotics, vasopressors, and mechanical ventilation. The policy can choose multiple treatments at the same time or no treatment at all, creating 8 different unique actions.

The simulator models patients as a combination of four vital signs: heart rate, blood pressure, oxygen concentration and glucose levels, all with discrete states (for example, for heart rate low, normal and high). There is a latent variable called diabetes that is present with a $20\%$ probability which drives the likelihood of fluctuating glucose levels. When a patient has at least 3 of the vital signs simultaneously out of the normal range, the patient dies. If all vital signs are within normal ranges and the treatments are all stopped, the patient is discharged. The reward function is $+1$ if a patient is discharged, $-1$ if a patient dies, and 0 otherwise.

We follow the process described by Oberst and Sontag (2019) to marginalize an optimal policy's action over 2 states: glucose level and whether the patient has diabetes. This creates the **Sepsis-POMDP** environment. We sample 200, 1000, and 5000 patients (trajectories) from Sepsis-POMDP environment with the optimal policy that has 5% chance of taking a random action. We also sample 200 trajectories from the original MDP using the same policy; we call this the **Sepsis-MDP** environment.

**Robomimic.** Our approach is further evaluated on a third domain, Robomimic (Mandlekar et al., 2021), consisting of various continuous control robotics environments along with corresponding sets of suboptimal human data. More specifically, we use the **Can-Paired** dataset composed of mixed-quality human data. These 200 demonstrations include an equal combination of "good" (the can is picked up and placed in the correct bin) and "bad" trajectories (the can is picked up and thrown out of the robot workspace). The initializations of the tasks being identical, it is expected that algorithms dealing with suboptimal data will be able to filter out the good trajectories from the bad ones and achieve near-optimal performance. Interestingly, state-of-the-art batch RL algorithms do not reach maximum performance (Mandlekar et al., 2021), making this task a good testbed for our procedure. We also use the **Transport** dataset, where two robot arms must transfer an object from one bin to another. The dataset contains 200 successful trajectories collected by one human operator.

**D4RL.** D4RL (Fu et al., 2020) is an offline RL standardized benchmark designed and commonly used to evaluate the progress of offline RL algorithms. We use 3 datasets of different quality from the Hopper task: hopper-random with 200k samples from a randomly initialized policy, hopper-medium with 200k samples from a policy trained to approximately 1/3 the performance of a policy trained to completion with SAC ("expert"), and hopper-medium-expert with 200k samples from a 50-50 split of medium and expert data. The Hopper task is to make a hopper with three joints, and four body parts hop forward as fast as possible.

### A.15 TutorBot Domain

We introduce a new TutorBot simulator that is designed to mimic 3-5th grade elementary school children in understanding the concept of calculating volume, and engaging them while doing so. We base certain aspects of this simulator on some experimental studies of this learning environment, where an RL policy can learn to teach. The state space includes children's pre-test score, anxiety level, thinking time, and whether it's the last question in the tutoring session. The action is to offer encouragement, a guided prompt, or a hint at each step of the tutoring.

The dynamics of TutorBot is a 4th-order Markov transition function that takes in anxiety and the amount of thinking time and updates a latent parameter that captures learning progress. For each simulated student learning trajectory, we pre-determine how many times this student will interact with the TutorBot. We denote this as $T$, which is the trajectory length. We calculated the relationship between $T$ and the pre-test score based on the aforementioned experimental study.

$$T = \text{round}(7 - 0.46 * \text{pre-test} + l), l \sim U([-1, 2])$$

$$\theta_x = [0, -0.05, -0.2, -0.5], \theta_h = [0.5, 0.3, 0.2, 0]$$

$$T(s_{t+1}|s_t, a_t) = \left[\text{pre-test}, [s_{t-3}, s_{t-2}, s_{t-1}, s_t]\theta_x^T, [s_{t-3}, s_{t-2}, s_{t-1}, s_t]\theta_h^T, \mathbb{1}\{t + 1 = T\}\right]$$

| TutorBot | Dimension | Description |
|---|---|---|
| State | 4 | Pre-test $\in \{0, 1, ..., 8\}$, Anxiety-level $\in [-1, 0]$ Thinking $\in [0, 1]+$, Pre-termination $\in \{0, 1\}$ |
| Action | 1 | 0 = Encourage, 1 = Guided Prompt, 2 = Hint |
| Reward | 1 | 0 for all steps if not last step |

Table A.7: MDP specification for TutorBot.

The reward is always 0 at all steps except for the final step. We use $x$ to denote anxiety and $h$ to denote thinking. Note that anxiety is always negative. We calculate the final reward as follows:

$$R_T = \mathbb{1}\{U[0, 1] < p\} * r_{\text{improv}} + (1 - \mathbb{1}\{U[0, 1] < p\}) * r_{\text{base}}, p = x + h$$

Under this simulator, a student will improve a small amount even if the chatbot fails to teach optimally.

$$r_{\text{improv}} \sim \mathcal{N}(\mu_{\text{improv}}, 1), r_{\text{base}} \sim \mathcal{N}(\mu_{\text{base}}, 0.4)$$

We provide the full simulator code in the GitHub repo.

## A.16 Sepsis-POMDP and Sepsis-MDP Experiment Details

Our algorithm-hyperparameter search is trying to be as realistic as possible to the setting of offline RL practitioners. We search over hyperparameters that could potentially have a strong influence on the downstream performance. Since this is an offline RL setting, we are particularly interested in searching over hyperparameters that have an influence on how pessimistic/conservative the algorithm should be.

### A.16.1 BCQ

Batch Constrained Q-Learning (BCQ) is a commonly used algorithm for batch (offline) reinforcement learning (Fujimoto et al., 2019). We search over the following hyperparameters:

| BCQ | Hyperparameter Range |
|---|---|
| Actor/Critic network dimension | [32, 64, 128] |
| Training Epochs | [15, 20, 25] |
| BCQ Threshold $\delta$ | [0.1, 0.3, 0.5] |

Table A.8: BCQ Hyperparams for Spesis-POMDP N=200, 1000. Sepsis-MDP N=200. TutorBot N=200.

BCQ threshold determines if the Q-network can take the max over action to update its value using $(s, a)$ – it can only update Q-function using $(s, a)$ if $\mu(s) > \delta$ and $\pi(a|s) > 0$. The higher $\delta$ (BCQ threshold) is, the less data BCQ can learn from. $\delta$ determines whether $(s', a') \in \mathcal{B}$.

$$Q(s, a) \leftarrow (1 - \alpha)Q(s, a) + \alpha(r + \gamma \max_{a' \text{s.t.}(s',a') \in \mathcal{B}} Q'(s', a')) \tag{8}$$

We search through the cross-product of these, in total 27 combinations.

For Sepsis-POMDP N=5000, we realize the network size is too small to fit a relatively large dataset of 5000 patients. So we additionally search over Table A.9. The actor/critic network uses a 2-layer fully connected network. This resulted in 6 additional combinations for BCQ in Sepsis-POMDP N=5000.

| BCQ | Hyperparameter Range |
|---|---|
| Actor/Critic network dimension | [256, 256], [512,512], [1024,1024] |
| Training Epochs | [25] |
| VAE Latent Dim | [512] |
| BCQ Threshold $\delta$ | [0.3, 0.4] |

Table A.9: BCQ Hyperparams for Spesis-POMDP N=5000.

## A.16.2 MBS-QI

The MBS-QI algorithm is very similar to BCQ, but MBS-QI also clips the states (Liu et al., 2020). We searched through similar hyperparameters as BCQ.

| MBS-QI | Hyperparameter Range |
|---|---|
| Actor/Critic network dimension | [32, 64, 128] |
| Training Epochs | [15, 20, 25] |
| BCQ Threshold $\delta$ | [0.1, 0.3, 0.5] |
| Beta $\beta$ | [1.0, 2.0, 4.0] |

Table A.10: MBS-QI Hyperparams for Spesis-POMDP N=200, 1000. Sepsis-MDP N=200. TutorBot N=200.

The beta ($\beta$) hyperparameter in MBS-QI is a threshold for the VAE model's reconstruction loss. When the reconstruction loss of the next state is larger than beta, MBS-QI will not apply the Q function on this next state to compute future reward (to avoid function approximation over unfamiliar state space).

$$
\begin{aligned}
\zeta(s, a; \widehat{\mu}, b) &= \mathbb{1}(\widehat{\mu}(s, a) \geq \beta) \\
(\tilde{\mathcal{T}} f)(s, a) &:= r(s, a) + \gamma \mathbb{E}_{s'}[\max_{a'} \zeta \circ f(s', a')]
\end{aligned}
\tag{9}
$$

We search through the cross-product of these, in total 81 combinations. Similar to BCQ situation, we realize the network size is too small to fit a relatively large dataset of Sepsis-POMDP N=5000. So we additionally search over Table A.11. The actor/critic network uses a 2-layer fully connected network. This results in 18 additional combinations for MBS-QI in Sepsis-POMDP N=5000.

## A.16.3 MOPO

We also experiment with Model-based Offline Policy Optimization (MOPO) (Yu et al., 2020). The original MOPO paper only experimented on Mujoco-based locomotion continuous control tasks. We want to experiment with whether MOPO can work well in environments like the Sepsis-POMDP simulator, which is not only a healthcare domain but also partially observable with a discrete state and action space. We do not expect MOPO to do well. We re-implemented two versions of MOPO

| MBS-QI | Hyperparameter Range |
|---|---|
| Actor/Critic network dimension | [256, 256], [512,512], [1024,1024] |
| Training Epochs | [25] |
| VAE Latent Dim | [512] |
| BCQ Threshold $\delta$ | [0.3, 0.4] |
| Beta $\zeta$ | [1.0, 2.0, 4.0] |

Table A.11: MBS-QI Hyperparams for Spesis-POMDP N=5000.

with Tensorflow 2.0 and PyTorch, and used the PyTorch version to run our experiments. Our implementation of MOPO matches the original's performance in a toy environment.

MOPO is fairly slow to run – because it needs first to train a model to approximate the original environment, and then sample from this model to train an RL algorithm. We did not evaluate it for Sepsis-POMDP N=5000.

| MOPO | Hyperparameter Range |
|---|---|
| Actor/Critic network dimension dim | [32, 32], [64, 64], [128, 128] |
| Training Iterations | [1000, 2000, 3000] |
| MOPO Lambda $\lambda$ | [0, 0.1, 0.2] |
| Number of Ensembles | [3, 4, 5] |

Table A.12: MOPO hyperparameters for Spesis-POMDP N=200, 1000.

Number of ensembles refers to MOPO Algorithm 2, which trains an ensemble of $N$ probabilistic dynamics on batch data. $N$ should be adjusted according to the dataset size. Each dynamics model is trained on $\frac{1}{N}$ of the data during each epoch.

$$\widehat{T}_i(s', r | s, a) = \mathcal{N}(\mu_i(s, a), \Sigma_i(s, a)) \tag{10}$$

MOPO $\lambda$ hyperparameter controls how small we want the reward to be, adjusting for state-action pair uncertainty. Generally, the more uncertain we are about $(s, a)$, the more we should ignore the reward that's outputted by the learned MDP model. Its use is also described in Algorithm 2:

$$\tilde{r}(s, a) := r(s, a) - \lambda \max_{i=1}^{N} ||\Sigma_i(s, a)||_F \tag{11}$$

We search through the cross-product of these, in total 81 combinations.

In our initial experiments, MOPO does not seem to perform well in a tabular setting where both state and action are discrete. Therefore, we simplified the idea of MOPO to introduce Pessimistic Ensemble MDP (P-MDP).

### A.16.4 P-MDP

As noted in the previous section, inspired by MOPO and MoREL (Kidambi et al., 2020), we develop a tabular version of MOPO. We instantiate $N$ tabular MDP models. For each epoch, each MDP model only updates on $1/N$ portion of the data. During policy learning time, for each timestep, we randomly sample 1 of the $N$ MDP for the next state and reward; and use Hoeffding bound to compute a pessimistic reward, similar to MOPO's variance penalty on reward:

Let $N(s, a)$ be the number of times $(s, a)$ is observed in the dataset:

$$\epsilon = \beta * \sqrt{\frac{2 \log(1/\delta)}{N(s, a)}} \tag{12}$$
$$\tilde{r}(s, a) = \min(\max(r - \epsilon, -1), 1)$$

In the last step we bound the reward to (-1, 1) for the Sepsis setting – but it can be changed to apply to any kind of reward range. We note that Hoeffding bound is often loose when $N(s, a)$ is small, therefore, might make the reward too small to learn any good policy. However, empirically, we observe that in the Sepsis-POMDP, P-MDP is often the best-performing algorithm. We additional add a temperature hyperparameter $\alpha$, that changes the peakness/flatness of the softmax distribution of the learned policy:

| P-MDP | Hyperparameter Range |
|---|---|
| Training Iterations | [1000, 5000, 10000] |
| Penalty Coefficient $\beta$ | [0, 0.1, 0.5] |
| Number of Ensembles | [3, 5, 7] |
| Temperature $\alpha$ | [0.05, 0.1, 0.2] |

Table A.13: P-MDP Hyperparams for Spesis-POMDP N=200, 1000.

Not surprisingly, since planning algorithms (such as Value Iteration or Policy Iteration) need to enumerate through the entire state space, we find it too slow to train a policy in Sepsis-MDP domain, because Sepsis-POMDP has 144 unique states, yet Sepsis-MDP has 1440 unique states (glucose level has 5 unique states and diabetes status has 2 unique states). TutorBot and Robomimic both have continuous state space, therefore are not suitable for our P-MDP algorithm without binning.

We search through the cross-product of these, in total 81 combinations.

For Sepsis-POMDP N=5000, we realize we can increase the number of MDPs and increase training iterations to fit a relatively large dataset of 5000 patients. So we additionally search over Table A.14. This results in 16 additional combinations for P-MDP in Sepsis-POMDP N=5000.

| P-MDP | Hyperparameter Range |
|---|---|
| Training Iterations | [20000, 40000] |
| Penalty Coefficient $\beta$ | [0.05, 0.1] |
| Number of Ensembles | [15, 25] |
| Temperature $\alpha$ | [0.01, 0.05] |

Table A.14: P-MDP Hyperparams for Spesis-POMDP N=5000.

### A.16.5 BC

Behavior Cloning (BC) is a type of imitation learning method where the policy is learned from a data set by training a policy to clone the actions in the data set. It can serve as a great initialization strategy for other direct policy search methods which we will discuss shortly.

One pessimistic hyperparameter we can introduce to behavior cloning is similar in spirit to BCQ and MBS-QI, we can train BC policy only on actions that the behavior policy has a high-enough probability to take, optimizing the following objective:

$$\zeta = \pi_b(a|s) \geq \alpha$$
$$\arg\min_\theta E_{(s,a)\sim D}||\pi_\theta(s) - \zeta \circ \pi_b(a|s)||^2 \quad (13)$$

We refer to $\alpha$ as the "safety-threshold". We search through the cross-product of these, in total 27 combinations.

| BC | Hyperparameter Range |
|---|---|
| Policy network dimension | [32, 32], [64, 64], [128, 128] |
| Training Epochs | [15, 20, 25] |
| Safety Threshold $\alpha$ | [0, 0.01, 0.05] |

Table A.15: BC Hyperparams for Spesis-POMDP N=200, 1000, 5000. Sepsis-MDP N=200. TutorBot N=200.

### A.16.6 POIS

Policy Optimization via Importance Sampling (Metelli et al., 2018) uses an importance sampling estimator as an end-to-end differentiable objective to directly optimize the parameters of a policy. In our experiment, we refer to this as the "**PG**" (policy gradient) method. Similar to BC method, we can set a safety threshold $\alpha$ that zeros out any behavior probability of an action that's not higher than $\alpha$, and then re-normalizes the probabilities of other actions. Metelli et al. (2018) also introduces another penalty hyperparameter $\lambda$ to control the effective sample size (ESS) penalty. ESS measures the Renyi-divergence between $\pi_b$ and $\pi_e$. Let $\widehat{V}$ be the differentiable importance sampling estimator – we write the optimization objective similar to Futoma et al. (2020), but without the generative model:

$$\mathcal{J}(\mathcal{D}_{\text{train}}) = \widehat{V}(\pi_\theta; \mathcal{D}_{\text{train}}) - \frac{\lambda_{\text{ESS}}}{\text{ESS}(\theta)}$$
$$\theta = \arg\max_\theta \mathcal{J}(\mathcal{D}_{\text{train}}) \quad (14)$$

We search through the following hyperparameters in Table A.16. There are 81 combinations in total.

### A.16.7 BC+POIS

BC + POIS is a method that first finds a policy using BC as an initialization strategy to make sure that the policy stayed close (at first) to the behavior policy. This is particularly useful for neural network-based policy classes, as a form of pre-training using behavior cloning objective. We use the same set of hyperparameters displayed in Table A.16, resulting in 81 combinations in total.

### A.16.8 BC+mini-POIS

In both Metelli et al. (2018) and Futoma et al. (2020), the loss is computed on the whole dataset $\mathcal{D}_{\text{train}}$, which makes sense – importance sampling computes the expected reward (which requires averaging over many trajectories to have an estimation with low variance). However, inspired by the success of randomized optimization algorithms such as mini-batch stochastic gradient descent

| BC | Hyperparameter Range |
|---|---|
| Policy network dimension | [32, 32], [64, 64], [128, 128] |
| Training Epochs | [15, 20, 25] |
| Safety Threshold $\alpha$ | [0, 0.01, 0.05] |
| ESS Penalty $\lambda$ | [0, 0.01, 0.05] |

Table A.16: POIS, BC+POIS, BC+mini-POIS Hyperparams for Spesis-POMDP N=200, 1000, 5000. Sepsis-MDP N=200. TutorBot N=200.

(SGD), we decided to attempt a version of BC + POIS with $\widehat{V}$ over a small batch of trajectories instead of over the entire dataset. Our batch size is 4 (4 trajectories/patients) for Sepsis-POMDP N=200 and 1000, which is very small. However, this strategy seems to be quite successful, resulting in learning high-performing policies competitive with other more principled methods. This can be seen in Figure A.2 ("BCMINIPG").

We leave the exploration of why this is particularly effective to future work, and hope others who want to try POIS style method to include our variant in their experiment. We use the same set of hyperparameters displayed in Table A.16, resulting in 81 combinations in total.

## A.17 TutorBot Experiment Details

The details of this environment is shown in the code file in the supplementary material. We trained BC+POIS, POIS, and BC+mini-POIS on this domain.

## A.18 Robomimic Experiment Details

We refer the reader to Mandlekar et al. (2021) for a full review of the offline RL algorithms used in our experiment. For Robomimic, we include the range of hyperparameters we have considered below:

- BC:
    - Actor NN dimension: [300,400], [1024,1024]
    - Training epochs: 600, 2000
    - GMM actions: 5, 25
- BCRNN:
    - RNN dimension: [100], [400]
    - Training epochs: 600, 2000
    - GMM actions: 5, 25
- BCQ:
    - Critic NN size: [300,400], [1024,1024]
    - Training epochs: 600, 2000
    - Action samples: [10,100], [100,1000]
- CQL:
    - Critic NN size: [300,400], [1024,1024]
    - Training epochs: 600, 2000
    - Lagrange threshold: 5, 25
- IRIS:
    - Critic NN size: [300,400], [1024,1024]
    - Training epochs: 600, 2000
    - LR critic: 0.001, 0.0001

## A.19 D4RL-Hopper Experiment Details

For the D4RL experiments, we include the range of hyperparameters we have considered below:

- BCQ:
    - Policy NN size: [512,512], [64,64]
    - LR policy: 0.001, 0.0001
- CQL:
    - Policy NN size: [256,256,256], [64,64,64]
    - LR policy: 0.001, 0.0001
- AWAC:
    - Policy NN size: [256,256,256,256], [64,64,64,64]
    - LR policy: 0.001, 0.0001

For BVFT Strategy 1 $\pi$ x FQE, we use the optimal FQE hyperparameter on all hyperparameters of BCQ, CQL and AWAC. For BVFT Strategy 2 $\pi$ + FQE, we use 4 FQE hyperparameters but only with 4 hyperparameters of BCQ. For RRS and CV, we use the optimal FQE hyperparameter on 4 hyperparameters of BCQ as well.

## A.20 Computing Resources

For the overall experimental study in this paper, an internal cluster consisting of 2 nodes with a total of 112 CPUs and 16 GPUs was used.

## A.21 Additional Offline RL Sensitivity Study

### A.21.1 Sensitivity to data splitting: One-Split OPE

Figure A.1 shows that procedure produces policies with drastically different estimated, and true, performances subject to randomness in data selection process. Because training and validation set randomness are directly conflated, it becomes difficult to accurately select a better `AH` pair (and its associated higher-performing policy) based on a single train/validation set partition.

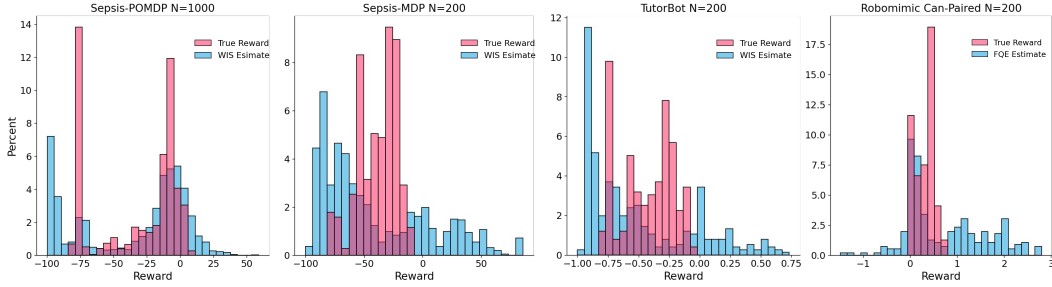

Figure A.1: We show that policies learned from offline RL algorithms are sensitive to the variation of training and validation dataset: an algorithm-hyperparameter (AH) pair can obtain wildly different policies based on which portion of the data they were trained on. We obtained 5400 policies from 540 AH combinations on Sepsis-POMDP (N=1000) domain. The variation is not just in terms of the policy's true performance in the real environment, but also in terms of OPE estimations. Note that FQE estimate on Robomimic exceeded the range of possible achievable rewards (between 0 and 1). The true reward is calculated by evaluating the policy in the real environment.

### A.21.2 Sensitivity to hyperparameters

In Figure A.2, we show that offline RL algorithms are sensitive to the choice of hyperparameters. In the Sepsis-POMDP N=1000 task and the Robomimic Can-Paired N=200 task, all popular offline RL algorithms show a wide range of performance differences even when trained on a fixed partition of the dataset.

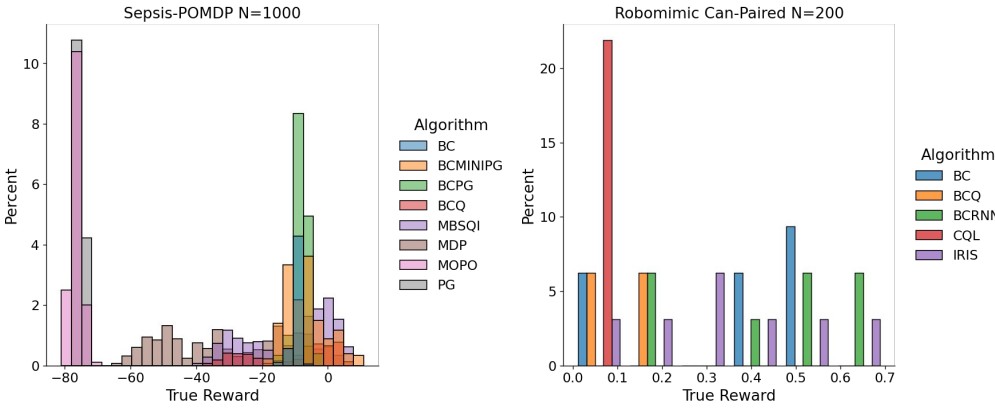

Figure A.2: Sensitivity of offline RL algorithms due to the choice of hyperparameters.