# OpenReview forum: "Data-Efficient Pipeline for Offline Reinforcement Learning with Limited Data"
_NeurIPS.cc/2022/Conference — NeurIPS 2022 Accept_

### Official Review · Reviewer_g82q · 2022-07-04

**Rating:** 7
**Confidence:** 4
**Soundness:** 4 excellent
**Presentation:** 4 excellent
**Contribution:** 4 excellent

**Summary:**

This paper proposes a workflow for hyperparameter and algorithm selection in offline RL. The authors argue that because datasets can be small and high variance, and algorithms themselves are hyperparameter sensitive, the prevailing approach of training on a fixed training set and tuning on a validation set is not sufficient. The authors instead argue to use re-sampling of the dataset and then average OPE estimates on the validation set in order to estimate the performance of the algorithm & hyperparameters. The authors evaluate on a collection of tasks (based on D4RL, MIMIC-III Sepsis, Robomimic, and Tutorbot) where they show their selection method typically finds the best policies compared to the baselines.

**Questions:**

- What was the computational cost of the various approaches used in evaluation? While SSR-RRS 5 produces the best results, does it also require significantly more compute that other approaches?
- My intuition is that OPE is fairly inaccurate in predicting the values of the policies, esp on high-dimensional domains. Was the accuracy of OPE a bottleneck for the method, or did it general predict the relative ordering of policies correctly?

**Limitations:**

The authors did not discuss potential negative societal impacts of their work, but they discussed a reasonable set of limitations for their method.

**Strengths And Weaknesses:**

- The evaluation is well-done and convincingly shows the utility of the proposed approach. The authors used several logical baselines and several domains that are fairly different qualitatively (ranging from healthcare to robotic control).
- This work has potential to be fairly significant within the field of offline RL, since it is applicable to most algorithms, and performs better than the common naive approach to hyperparameter selection.
- The writing is clear and well-organized, The authors motivate the use of their method from theory, describe their practical approach, and clearly explain the rationale behind their chosen baselines.

---

> ### Author Response · Authors · 2022-08-02
> **Author Response**
>
> Thank you for the thoughtful feedback and for appreciating the potential of our work having a significant impact in the offline RL community. Here is our response to address your questions and concerns.
>
> > ​​What was the computational cost of the various approaches used in evaluation? While SSR-RRS 5 produces the best results, does it also require significantly more compute that other approaches?
>
> Most of the approaches we evaluate leverage multiple repetitions (resampling) to account for data allocation randomness. We provide a time complexity table below:
>
> H = number of AH pairs to evaluate
>
> N = total data samples. We assume the training time for each trajectory is $N_1$ and evaluation time for each trajectory is $N_2$, where $N = N_1 + N_2$
>
> M = number of folds in multi-fold cross-validation
>
> B = number of bootstrap (this number is 100 in our experiment)
>
> P = number of resolutions for BVFT’s grid (proposed in [4])
>
> For BVFT, one can amortize the computational cost by caching (storing Q(s, a) for all (s, a) in the dataset). If caching is done only once, we treat the actual computation time for the validation data set as $n_2$. P is usually around 5-10.
>
> |                         | Training Complexity                     | Evaluation Complexity                |
> |-------------------------|-----------------------------------------|--------------------------------------|
> | One-Split               | H * $N_1$                               | H * $N_2$                            |
> | Bootstrapping (BCa)     | H * $N_1$                               | H * $B N_2$                          |
> | M-Fold Cross-Validation | H * M * $N * (M-1)/M$ = H * $N$ * (M-1) | H * M * $N$ * 1/M = H * N            |
> | K-Repeat RRS            | H * K * $N_1$                           | H * K * $N_2$                        |
> | BVFT                    | H * $N_1$                               | H * H * $N_2$ or H * H * $n$         |
> | BVFT-auto [4]           | H * $N_1$                               | P * H * H * $N_2$ or P * H * H * $n$ |
>
> When H is relatively large, for example, H = 540 (in our experiment), H * H = 2.916e5. It’s easy to see that RRS is slightly more expensive than M-Fold CV, but less expensive than the pairwise comparison tournament algorithm (BVFT). It’s also worth noting that BCa has a high evaluation cost when B is large – when B = 100, BCa evaluation cost is significantly higher than CV and RRS.
>
> > ​​My intuition is that OPE is fairly inaccurate in predicting the values of the policies, esp on high-dimensional domains. Was the accuracy of OPE a bottleneck for the method, or did it general predict the relative ordering of policies correctly?
>
> The choice of OPE does matter for our pipeline. Thankfully, heuristics for optimal OPE selection has been discussed by [4]. There have also been a growing number of works on automatic OPE selection [5][6]. Our contribution is orthogonal to these works and any future OPE method can be integrated into our pipeline.
>
> We want to thank you again for providing valuable feedback and reviewing our paper. Have we addressed your questions and comments? We are happy to discuss or answer more questions if they come up!
>
> [4] Voloshin, Cameron, et al. "Empirical study of off-policy policy evaluation for reinforcement learning." arXiv preprint arXiv:1911.06854 (2019).
>
> [5] Lee, J. N., Tucker, G., Nachum, O., and Dai, B. (2021). Model selection in batch policy optimization. arXiv preprint arXiv:2112.12320.
>
> [6] Su, Y., Srinath, P., and Krishnamurthy, A. (2020). Adaptive estimator selection for off-policy evaluation. In International Conference on Machine Learning, pages 9196–9205. PMLR

---

> > ### Comment · Reviewer_g82q · 2022-08-06
> > **Will maintain rating as accept**
> >
> > Thank you for the response, it answered most of my questions. I will keep my rating of accept as I think this paper is quite broadly applicable and will be of interest to the NeurIPS community.

---

### Official Review · Reviewer_CB5Q · 2022-07-04

**Rating:** 7
**Confidence:** 2
**Soundness:** 3 good
**Presentation:** 3 good
**Contribution:** 3 good

**Summary:**

The paper proposes SSR, a new pipeline for selecting the best-performing deployment policy in the offline RL setting. The key idea of SSR is the use of repeated random sampling (RRS), which samples $K$ overlapping train-test pairs from the offline dataset. Each specific algorithm-hyperparameter choice is trained and evaluated $K$ times using these pairs. SSR picks the policy with the highest average performance. The authors justify SSR both theoretically and empirically. SSR consistently outperforms other policy selection methods over various environments, especially in low-data regime.

**Questions:**

- Take computation aside, will increasing $K$ always improve the performance of the final policy?

Minor questions/suggestions:
- Legend of Figure 4(b) is confusing, is there a typo there?

**Limitations:**

The authors have adequately addressed the limitations of the work.

**Strengths And Weaknesses:**

### Strengths
- The paper tackles the problem of hyperparameter tuning, or policy selection, which is a relevant and essential aspect of offline RL.
- To the best of my knowledge, the idea presented is original. While repeated random sampling is not a new technique, it has never been tried in the offline RL setting.
- The idea is simple and well motivated. The authors did a good job of pointing out the main limitation of current OPE and OPL methods: the high variance with respect to the train-validation split. SSR naturally solves this issue by performing this split multiple times and averaging the performance.
- The authors evaluate SSR on extensive benchmarks, ranging from robotics to medical domains. SSR consistently produces better policies than other selection methods.
- The paper is easy to read in general.

### Weaknesses
- My first concern regards the robustness of SSR with respect to the chosen offline RL algorithm. In the paper, the authors only show the performance of the best policy among all AH pairs. However, it is also interesting to see if SSR consistently performs well given a specific algorithm. This reduces policy selection to hyperparameter tuning, which is important for future research in offline RL.
- The second concern regards the effectiveness of SSR with respect to the choice of OPE. Will there be an OPE that is robust enough to the partitioning of the dataset that using repeated sampling will not help? I would love to see at least one more OPE being used in SSR in addition to WIS or FQE.

---

> ### Author Response · Authors · 2022-08-02
> **Author Response**
>
> Thank you for your thoughtful questions and comments.
>
> > robustness of SSR with respect to the chosen offline RL algorithm. In the paper, the authors only show the performance of the best policy among all AH pairs. see if SSR consistently performs well given a specific algorithm.
>
> Thank you for this interesting suggestion. Here we show that SSR-RRS can still robustly select a good hyperparameter for a given offline RL policy learning algorithm (the gap between best AH selected and true best AH is relatively small).
>
> | Sepsis-POMDP | Range of True Policy Performance (95%CI) | Percentile of Selected AH |  Performance of Hyperparameter Chosen by SSR-RRS 5 | True Best hyperparameter  Performance |
> |--------------|------------------------------------------|---------------------------|----------------------------------------------------|---------------------------------------|
> | BCQ          | [-10.8, -0.73]                           | 94%                       | 5.98                                               | 7.86                                  |
> | MBSQI        | [-7.34, -2.26]                           | 95%                       | 6.40                                               | 7.42                                  |
> | BC           | [-8.98, -8.37]                           | 58%                       | -8.46                                              | -7.42                                 |
> | BC + PG      | [-5.55, -4.26]                           | 78%                       | -3.68                                              | 2.52                                  |
> | P-MDP        | [-31.17, -21.26]                         | 83%                       | 0.23                                               | 2.82                                  |
>
> Each algorithm we evaluate over 24 to 72 hyperparameters and we compute the 95% confidence interval of all these policies’ true performance. This shows that except for BC, we are picking hyperparameters that are out-performing 78%-95% of other hyperparameters in the same algorithm.
>
> > Will there be an OPE that is robust enough to the partitioning of the dataset that using repeated sampling will not help?
>
> This is a great question. We suspect that there does not exist an OPE for which a single train-test split would always suffice (aka no need for repeated sampling). This is because the estimates produced by off policy evaluation methods are inherently a function of the heterogeneity of the trajectories in the dataset itself and a single train-test partition, particularly in small datasets, may lead to erroneous conclusions as we show in Theorem 1.
> > I would love to see at least one more OPE being used in SSR in addition to WIS or FQE.
>
> We appreciate the good suggestionWe additionally evaluated clipped IS (importance sampling) and CWPDIS [1] as two additional OPE methods in one domain (please see the appendix in the updated version)
>
> | Sepsis-POMDP | Best AH Performance Chosen by SSR-RRS 5 |
> |--------------|-----------------------------------------|
> | WIS          | 6.75                                    |
> | IS           | 4.47                                    |
> | CWPDIS       | 4.68                                    |
> | FQE-1        | 2.84                                    |
>
> The choice of OPE does matter for our pipeline. Thankfully, heuristics for optimal OPE selection has been discussed by [2]. There have also been a growing number of works on automatic OPE selection [3][4]. Our contribution is orthogonal to these works and any future OPE method can be integrated into our pipeline.
> Have we addressed your questions and comments? We are happy to discuss or answer more questions if they come up.
>
> [1] Thomas, P., & Brunskill, E. (2016, June). Data-efficient off-policy policy evaluation for reinforcement learning. In International Conference on Machine Learning (pp. 2139-2148). PMLR.
>
> [2] Voloshin, Cameron, et al. "Empirical study of off-policy policy evaluation for reinforcement learning." arXiv preprint arXiv:1911.06854 (2019).
>
> [3] Lee, J. N., Tucker, G., Nachum, O., and Dai, B. (2021). Model selection in batch policy optimization. arXiv preprint arXiv:2112.12320.
>
> [4] Su, Y., Srinath, P., and Krishnamurthy, A. (2020). Adaptive estimator selection for off-policy evaluation. In International Conference on Machine Learning, pages 9196–9205. PMLR

---

> > ### Author Response · Authors · 2022-08-07
> > **Dear Reviewer**
> >
> > We hope our additional experiments and explanations have addressed the concerns raised in the review. Since there is a limited discussion window, we are eager to learn whether you have further questions. Thanks again for your valuable suggestions and comments.

---

> > > ### Comment · Reviewer_CB5Q · 2022-08-08
> > > **Increasing my score**
> > >
> > > I thank the authors for the detailed response and the additional experiments. The response addressed my concerns, therefore I'm raising my score to an Accept.

---

### Official Review · Reviewer_5T2Y · 2022-07-10

**Rating:** 6
**Confidence:** 3
**Soundness:** 2 fair
**Presentation:** 3 good
**Contribution:** 2 fair

**Summary:**

This paper studies the model and hyper-parameter selection in offline RL, which is an important problem as most offline RL algorithms are sensitive to the algorithm it used, the hyper-parameters and other subtle algorithm details. Different from supervised learning, OPE estimate on the validation set is not reliable when the sample size is small and the trajectory is long. This paper proposes SSR (Repeated Random Sampling) for algorithm selection, the idea is simple in which we just randomly split the trajectories as train/val multiple times, and perform algorithm training and OPE on the train/val separately. The algorithm configuration that achieves the highest average performance on the val will be selected, and finally re-train the whole dataset using this selected configuration. Empirical studies on different domains, such as healthcare, robotics show better performance compared with other baselines, such as BVFT, CV.

**Questions:**

[1]. I appreciate the effort on making the pipeline algorithm agnostic. One thing for further understanding this, does the author observe any correlation about the training algorithm used and the OPE used for selection? Say is FQE performs better on FQI algorithms in terms of selection compared with WIS? Is there any interplay on this?

[2]. The empirical study tries to span different domains, which is very nice. It would be very helpful if the authors try to organize them in a way that shows the properties of these MDP/POMDP problems, such as sto/det reward, transitions, sparse/dense reward, horizon length etc and the dataset composition: narrow or diverse. It might be interesting to see how the method works across these different settings, and provides some practical recommendations

[3]. Any commends/ideas on reducing the time complexity of the algorithm, as it involves training for multiple times.


Minor:

[1]. What are the y-axis in Fig 4(a)? It seems we have 540 AH pairs, but the y-axis is much smaller than this.
[2]. Is there a typo on the legend of Fig 4(b)? SSR-RRS 5 is missing.

**Ethics Review Area:**

["I don’t know"]

**Limitations:**

Yes.

**Strengths And Weaknesses:**

Strength:

[1]. The paper takes a step towards an important problem in offline RL, i.e., the hyper-parameter selection. As most offline RL algorithms are very sensitive to this and it is important to understand the rule/pipeline for how to effectively select the best configuration for the training algorithm.

[2]. The method proposes is simple and well-motivated, especially on how it compared with K-fold CV.

[3]. The ablation study is well-done, especially on the importance of re-train procedure, the impact of the repeats training and the effect of the datasize.

Weakness:

[1]. This method somehow sounds like more of a heuristic. Though Theorem 1 tries to show there exists a setting that this k-split average performance is better than the 1 split, it does not show the correctness of the method, and this result seems weak for the theoretical justification.

[2]. If I understand it correctly, the computational burden for the method seems high, as it needs to retrain for K times, and the training for RL is very time-consuming.

[3]. The paper frames it as an overall pipeline, it discusses more on train/val split, however for the OPE estimate part is missing. How is the pipeline sensitive to the OPE estimate used? Is there any ablation study on this? In the experiments, it just uses the default setting, WIS for discrete and FQE for continuous task.

---

> ### Author Response · Authors · 2022-08-02
> **Author Response (1)**
>
> Thank you for your thoughtful questions and comments.
>
> > Theorem 1… does not show the correctness of the method, and this result seems weak for the theoretical justification.
>
> Thank you for highlighting this issue. Indeed though such results are common in supervised learning, the offline RL literature is largely lacking (to our knowledge) formal bounds on the ability to select the optimal algorithm and policy in the finite sample regime, without additional assumptions like a Markov environment, realizability, completeness and data coverage. Our contribution is to highlight a previously little discussed issue, the impact of dataset partitioning, which is often used by policy selection and learning methods that make minimal assumptions on the domain and dataset.
>
> Our RSS is a finite approximation of Leave-p-out (Lp0) cross-validation [1]. LpO is known in supervised learning, but rarely used due to the computational burden. The correctness of LpO is proved in [2] in a supervised learning setting with projection estimators. Unlike K-fold cross-validation, Leave-p-out CV selects p data points for evaluation and the rest for training. In our proposed RSS method, we set $p = n / 2$, and instead of exhaustively enumerating all possible selection of p data points out of n data points, we only repeat this process K times. Asymptotically as the amount of data goes to infinity, this approach should be correct, but also a single train/test split will also be correct in such a setting. The key challenges arise in the finite data setting, where the choice of dataset partitioning is key. We present Theorem 1 to illustrate the importance of repeated experiments and the potential failure of selecting an AH with a single train/test split when the dataset is limited. We have updated our text to include this discussion (in the appendix).
>
> [1] https://scikit-learn.org/stable/modules/generated/sklearn.model_selection.LeavePOut.html
>
> [2] Celisse, Alain. "Optimal cross-validation in density estimation with the $ L^{2} $-loss." The Annals of Statistics 42.5 (2014): 1879-1910.
>
> > [2]. If I understand it correctly, the computational burden for the method seems high, as it needs to retrain for K times, and the training for RL is very time-consuming.
>
> We completely agree the computational method is large. Unfortunately larger cost for selecting hyperparameters or models is often common: in supervised learning, cross validation is a common approach, which will have a similar cost as our RSS– CV will require recomputing a policy for each fold and algorithm/hyperparameters. When the data is finite and it is costly to deploy or gather data (such as health care settings, or student education), additional extra training time to increase the chance of obtaining a high performing policy for future use is likely to be worth the benefit. We have added this point more directly in the paper and also added a time complexity to the appendix, which highlights and compares the computational cost.
>
> > [3]. The paper frames it as an overall pipeline, it discusses more on train/val split, however for the OPE estimate part is missing. How is the pipeline sensitive to the OPE estimate used? Is there any ablation study on this?
>
> We appreciate the good suggestion. Indeed, as has motivated much of the literature on OPE for RL, the OPE method chosen will impact the resulting performance of our method. We expect that minimal-assumption OPE methods like weighted doubly robust methods (e.g. Jiang and Li 2015; Thomas and Brunskill 2016) may be most consistently accurate for many domains (as shown in a recent face off paper [4]), though if the domain is known to be Markov and the models are well specified, FQE methods will be more data efficient.
>
> There exist several recent methods for automatic OPE selection (e.g. [5][6]). Our pipeline can be used jointly with these methods and any future method that proposes better automatic OPE selection.
>
> Building on the reviewer’s suggestion to show the impact, in the table below we show using our RSS procedure with both WIS and with 8 different FQE OPE methods. Note for the below, that the OPE method FQE assumes the domain is Markov, which Sepsis-POMDP is not, so it is not so surprising that using a poor fit for the domain OPE method will result in worse selection.
>
> (to be continued)

---

> > ### Author Response · Authors · 2022-08-02
> > **Author Response (2)**
> >
> > (following from above)
> >
> > | Sepsis-POMDP | Parameters               | RRS  Final Selected Policy’s Performance |
> > |--------------|--------------------------|------------------------------------------|
> > | FQE-1        | [64], lr=3e-4, epoch=20  | 2.84                                     |
> > | FQE-2        | [64], lr=1e-5, epoch=20  | -74.26                                   |
> > | FQE-3        | [64], lr=3e-4, epoch=50  | -20.88                                   |
> > | FQE-4        | [64], lr=1e-5, epoch=50  | -14.16                                   |
> > | FQE-5        | [128], lr=3e-4, epoch=20 | -75.26                                   |
> > | FQE-6        | [128], lr=1e-5, epoch=20 | -14.48                                   |
> > | FQE-7        | [128], lr=3e-4, epoch=50 | -75.54                                   |
> > | FQE-8        | [128], lr=1e-5, epoch=50 | -74.26                                   |
> > | IS           | N/A                      | 4.47                                     |
> > | CWPDIS [7]   | N/A                      | 4.68                                     |
> > | WIS          | N/A                      | 6.75                                     |
> >
> > We also wish to highlight that it is well known that OPE methods are important, but our paper is focused on an under-explored issue: that the dataset partitioning can also introduce a substantial amount of *additional* impact on learning good policies / selecting good hyperparameters/algorithms, as we demonstrated in the paper.
> >
> >
> > [4] Voloshin, C., Le, H. M., Jiang, N. and Yue, Y. (2019). Empirical study of off-policy policy evaluation for reinforcement learning. arXiv preprint arXiv:1911.06854.
> >
> > [5] Lee, J. N., Tucker, G., Nachum, O., and Dai, B. (2021). Model selection in batch policy optimization. arXiv preprint arXiv:2112.12320.
> >
> > [6] Su, Y., Srinath, P., and Krishnamurthy, A. (2020). Adaptive estimator selection for off-policy evaluation. In International Conference on Machine Learning, pages 9196–9205. PMLR
> >
> > [7] Thomas, P., & Brunskill, E. (2016, June). Data-efficient off-policy policy evaluation for reinforcement learning. In International Conference on Machine Learning (pp. 2139-2148). PMLR.
> >
> > > [1]. I appreciate the effort on making the pipeline algorithm agnostic. One thing for further understanding this, does the author observe any correlation about the training algorithm used and the OPE used for selection? Say is FQE performs better on FQI algorithms in terms of selection compared with WIS? Is there any interplay on this?
> >
> > Thanks for the interesting question. In our evaluation on Sepsis, FQE is used to evaluate both BCQ and MBSQI (both FQI-based) and BC and BCPG (policy-gradient algorithms).
> >
> > We designed the following analysis experiment using our logged results. We first rank all AH pairs (540 of them) with their true performance in the simulator, and then we count the percentage of FQI (BCQ, MBSQI) algorithms that appear in the top-10%, 20%, and 50% percentile. The number in each cell should be read as: “90.7% of AH pairs in the top-10% based on True Performance are FQI-based”. If FQE is biased towards FQI algorithms, then we expect to see a higher percentage of BCQ and MBSQI AH pairs selected than the true performance baseline and to other OPE methods.
> >
> > | Sepsis-POMDP OPE Method | % of BCQ and MBSQI AHs  in Top-10% AHs | % of BCQ and MBSQI AHs  in Top-20% AHs |
> > |-------------------------|----------------------------------------|----------------------------------------|
> > | True Performance        | 90.7%                                  | 61.1%                                  |
> > | FQE-1                   | 0%                                     | 0%                                     |
> > | WIS                     | 9.4%                                   | 35.5%                                  |
> > | RRS-5 WIS               | 68.5%                                  | 58.3%                                  |
> >
> > Based on this analysis, we can see that FQE is not biased to select FQI-based algorithms in the Sepsis-POMDP domain. However, our analysis is limited to one domain and we will include this analysis in our appendix.

---

> > > ### Author Response · Authors · 2022-08-02
> > > **Author Response (3)**
> > >
> > > > [2]. The empirical study tries to span different domains, which is very nice. It would be very helpful if the authors try to organize them in a way that shows the properties of these MDP/POMDP problems, such as sto/det reward, transitions, sparse/dense reward, horizon length etc and the dataset composition: narrow or diverse. It might be interesting to see how the method works across these different settings, and provides some practical recommendations
> > >
> > > Thank you for this great suggestion. We actually had included some properties of each dataset in Table A2 in the appendix in terms of Average Trajectory Length, MDP/POMDP, and Number of Trajectories in the dataset. We have now included an additional table that provides information that you proposed. We recommend using a higher number of K for smaller datasets.
> > >
> > > > [3]. Any commends/ideas on reducing the time complexity of the algorithm, as it involves training for multiple times.
> > >
> > > Thank you for asking about this. As we suggest above, the popularity of cross validation in supervised learning for smaller datasets for model/hyperparameter selection indicates that some repeated training may be important. Perhaps some strategic data selection method might make each repetition more informative, such as identifying high reward trajectories and distributing them equally among training and evaluation partitions. We leave such investigation to future work.
> > >
> > > > [1]. What are the y-axis in Fig 4(a)? It seems we have 540 AH pairs, but the y-axis is much smaller than this. [2]. Is there a typo on the legend of Fig 4(b)? SSR-RRS 5 is missing.
> > >
> > > Thank you for catching these – we have fixed these issues. One of the conditions is SSR-RRS 5.
> > > Have we addressed your questions and comments appropriately? We are happy to discuss or answer more questions if they come up.

---

> > > > ### Author Response · Authors · 2022-08-07
> > > > **Dear Reviewer**
> > > >
> > > > We hope our additional experiments and explanations have addressed the concerns raised in the review. Since there is a limited discussion window, we are eager to learn whether you have further questions. Thanks again for your valuable suggestions and comments.

---

> > > > > ### Comment · Reviewer_5T2Y · 2022-08-09
> > > > > **Thanks for the author response!**
> > > > >
> > > > > Thanks for the author response and the additional effort on running the experiments of different OPEs. Though i am still a little bit concerned about the computationally complexity (which might be an interesting future work), I am willing to adjust my score.

---

### Official Review · Reviewer_DpHr · 2022-07-10

**Rating:** 8
**Confidence:** 4
**Soundness:** 4 excellent
**Presentation:** 4 excellent
**Contribution:** 3 good

**Summary:**

The paper proposes a method for offline policy selection for offline reinforcement learning algorithms. The main idea is to use repeated random splitting of the dataset to pick the best hyper-parameters before training on the entire dataset. The algorithm is evaluated against multiple datasets and compared against standard benchmarks.

**Questions:**

Q1. The proof of theorem 1 assumes that the behavior policy is uniform random, and that high reward trajectories are sparse. This may not be true in practice. Historical data is usually from a non-random policy already deployed in a system. Can you explain why this assumption holds, and what are the implications if it is voided.

Q2. In line 174, you claim "WIS will return the observed return of the behavior policy if averaging over a single trajectory, independent of the target policy to be evaluated." It is not clear why this is true. There is no evidence, analysis or citation.

Q3. Line 207 claims: "As K → ∞, RRS approaches the leave-p-out cross-validation (CV), where p denotes the number of examples in the validation dataset." Again, not clear why this is true. How does p relate to the size of the dataset you use in RRS (split by 2)?

**Limitations:**

The entire paper relies on a single evaluation method for a dataset (WIS for discrete, FQE for continuous MDPs). It would be good to see if the claims hold for another evaluation method.

**Strengths And Weaknesses:**

Strengths:
- The paper tackles an important problem in offline reinforcement learning. Without a policy selection method, it is likely that the resulting policies perform poorly after deployment.
- The algorithm proposed is well motivated with theory and compared with the related works in literature.
- The evaluation procedure is systematic, with datasets from multiple domains in reinforcement learning and comparisons against the state-of-the-art policy selection algorithms.

Weaknesses:
- Some of the claims in the paper do not have evidence or reasoning. Details below.

---

> ### Author Response · Authors · 2022-08-02
> **Author Response**
>
> Thank you for the thoughtful feedback and appreciating the importance of policy selection in the offline RL setting. We believe that previous works do not investigate the limitation of sharing a single dataset for both algorithm training and evaluation and our work will provide a practical guideline on how to deploy offline RL algorithms for the limited data setting.
>
> We appreciate all the questions you raised. Here are our explanations:
>
> > Q1. The proof of theorem 1 assumes that the behavior policy is uniform random, and that high reward trajectories are sparse. This may not be true in practice. Historical data is usually from a non-random policy already deployed in a system. Can you explain why this assumption holds, and what are the implications if it is voided.
>
> Thank you for raising this concern. It is true that a behavior policy with a certain performance is usually deployed to collect data that will certainly outperform a uniform random policy.
>
> > Q2. In line 174, you claim "WIS will return the observed return of the behavior policy if averaging over a single trajectory, independent of the target policy to be evaluated." It is not clear why this is true. There is no evidence, analysis or citation.
>
> We apologize for not initially providing support for this statement. In brief, WIS works by first computing the probability of the dataset trajectory appearing under the evaluation policy and behavior policy:
> $$ w_i = \prod^L_{t=1} \frac{\pi_e(a_t \vert s_t)}{\pi_b(a_t \vert s_t)} $$
> Then, this coefficient is normalized before multiplying with the trajectory return, therefore:
> $$\text{WIS}(D) = \frac{1}{n} \sum_{i=1}^n \frac{w_i}{\sum_{j=1}^n w_j} (\sum_{t=1}^L \gamma^t R_t^i) $$
> which means that, perhaps surprisingly, if there is a single trajectory, $n=1$:
> $$\text{WIS}(D) =\frac{w_i}{w_i} (\sum_{t=1}^L \gamma^t R_t^i) = \sum_{t=1}^L \gamma^t R_t^i $$
> then we can see that WIS is a biased estimator that returns just the reward obtained from that trajectory, regardless of $w_i$ (which captures the distributional difference between evaluation policy and behavior policy). A more detailed explanation can be found in [1]. We use this example to show that a popular non-parametric OPE technique is heavily influenced by other data points in the evaluation dataset.
>
> > Q3. Line 207 claims: "As K → ∞, RRS approaches the leave-p-out cross-validation (CV), where p denotes the number of examples in the validation dataset." Again, not clear why this is true. How does p relate to the size of the dataset you use in RRS (split by 2)?
>
> We apologize for not explaining this more clearly. “Leave-p-out cross-validation” refers to leave **p** data points out (i.e., “leave-one-out cross-validation” means leave 1 data point for evaluation and use the rest for training). Leave-p-out cross-validation however, need to consider the “selection” of p points when $p > 1$. Which two points should be left out for evaluation? Therefore, leave-p-out cv will generate (n choose p) partitions and compute average over all these partitions. It’s easy to see how this number will grow very large when $p > 1$ if $n$ is large. In RSS, since each time, we randomly choose half of the dataset to be used for training and the rest for evaluation, we are setting $p = n / 2$, and if we repeat RSS infinitely many times, we will enumerate through all possible combinations that leave-p-out cv will have to go through when $p = n / 2$. Therefore, RSS can be treated as an approximation (but not exact) version of leave-p-out cross-validation under a finite computation budget.
>
> Have we addressed your questions and comments appropriately? We are happy to discuss or answer more questions if they come up!
> Reference:
>
> [1] Monte Carlo Sampling https://artowen.su.domains/mc/Ch-var-is.pdf (Page 8 – Self-normalized importance sampling)

---

> > ### Author Response · Authors · 2022-08-07
> > **Dear Reviewer**
> >
> > We hope our explanations are satisfactory to you. We are happy to answer more questions or clarify any points if you have them! Thanks again for your valuable suggestions and comments.

---

> > > ### Comment · Reviewer_DpHr · 2022-08-07
> > > **Rebuttal response**
> > >
> > > Thank you. Yes, your responses adequately answered my questions. Please consider including these in the paper.

---

### Official Review · Reviewer_x8P4 · 2022-07-16

**Rating:** 7
**Confidence:** 5
**Soundness:** 3 good
**Presentation:** 3 good
**Contribution:** 3 good

**Summary:**

This paper intends to improve offline reinforcement learning (RL) with limited data from a data-partition perspective. The authors first analyze why traditional data splittings don't perform well, including non-splitting (on the entire dataset) and cross-validation splitting. Finally, the authors propose a repeating random sampling strategy to enhance the selection of algorithm-hyperparameter (AH). The authors design experiments to demonstrate the superiority of the proposed SSR-RRS method in terms of policy performance.

**Questions:**

1. In line 37, the authors claim that the trained policies cannot be well evaluated, when the validation set has limited high-reward trajectories. This is doubtful, because the training set is responsible for the learning process, while the validation set is only used to 'predict'. It seems that after the policy is trained on the training set, its 'prediction' capability is not affected by the validation set. Is there any justification for that?
2. In RRS, can the policies trained by previous splittings be used as the pre-training for current splitting?

**Limitations:**

There is no apparent limitation on the work. It is universal and can be adapted to different offline RL algorithms. It will have a broad impact on the RL area.

**Strengths And Weaknesses:**

Strengths:
1. The research topic, improving offline RL from the data splitting angle, is inspiring and innovative. Data splitting is widely investigated in supervised learning, but it is rarely studied in the RL area. This paper proves that data splitting has a significant impact on offline RL training.
2. The idea of repeated random sub-sampling (RRS) is novel. The RRS strategy integrates ideas of ensemble learning and cross-validation, which is maybe not very fancy, yet very innovative and effective.
3. Almost all claims and theorems are justified either from the theoretical or the empirical analysis, making the paper solid and sound.


Weaknesses:
1. The parameter sensitive analysis for the RRS method is not comprehensive. To make the work more practical, the readers would like to know how to choose a reasonable splitting number $K$. The experiments show that when $K=5$ has the best performance. Does this mean the larger $K$ the better? Will it drop when reaching some thresholds?

---

> ### Author Response · Authors · 2022-08-02
> **Author Response**
>
> We would like to thank the reviewer for their thoughtful review and the positive feedback that our intention to investigate offline RL algorithm from data splitting angle is inspiring and innovative, and both of our theoretical analysis and empirical result are sound and solid. We will make sure to incorporate all the suggestions.
> > parameter sensitive analysis for the RRS method
>
> Thank you for bringing up this point. In general we expect the issue of data partitioning into a train and test split is most important in small datasets: as the dataset gets very large, a single train/test split will generally work well. Therefore, we suggest using a larger K for smaller datasets, but for larger datasets a smaller K will likely be sufficient. Using our theoretical example in the appendix (chain-MDP), this can also be observed – with larger N, the failure probability for smaller numbers of repeats decreases. This N-K tradeoff has computational benefits if there is a limited computational budget (larger datasets will require more training, therefore it is harder to use a larger K).
>
> > In line 37… claim that the trained policies cannot be well evaluated, when the validation set has limited high-reward trajectories. This is doubtful, because the training set is responsible for the learning process … after the policy is trained on the training set, its 'prediction' capability is not affected by the validation set.
>
> Perhaps surprisingly, unlike supervised learning, the validation set composition can have a large impact on the estimation quality of off policy evaluation. Due to the potential difference in the behavior policy used to gather data, and the evaluation policy of interest, when the validation set has limited overlap with the evaluation policy (such as when the evaluation policy would reach high reward trajectories much more frequently than those are present in the available data), the estimate of the quality of the evaluation policy can suffer. We provide an example of this in our proof of how a single train/test split may fail though repeated sampling succeeds: please see appendix A.2.
>
> > In RRS, can the policies trained by previous splittings be used as the pre-training for current splitting?
>
> Thank you for this suggestion. While we agree pretraining might speed the policy learning procedure, we find, consistent with much of offline RL work, that it is beneficial to use random restarts for the current splitting. This is because the optimization process is not convex and the randomness helps us explore other possible policies.
>
> Did this answer your questions? We are also happy to answer more questions if they arise.

---

> > ### Comment · Reviewer_x8P4 · 2022-08-03
> > **I will keep the rating as 'Accept'**
> >
> > Thanks for the detailed response. It answered most of my questions.

---

### Meta-Review · Area_Chair_SNzD · 2022-08-29

**Recommendation:** Accept
**Confidence:** Certain

**Metareview:**

## Summary
In offline RL, typically it is not possible to run the policies to evaluate them in the environment for model selection, as a result one needs to rely on the offline approaches such as OPE methods to do the model selection. This paper points to the importance of data partitioning over single partitioning for offline model and hyper parameter selection. They propose a simple but effective method called SSR for this purpose and compare non-splitting and different ways of cross-validation splitting. The paper has comprehensive experiments on D4RL, Robomimic, Tutorbot and Sepsis domains. They show very promising results on those tasks over other approaches for offline policy selection.

## Decision
Overall this paper is very well-written and studying a very important problem for offline RL. The paper was already in a good shape when submitted, however, after a few small points made by the reviewers I think it will be in a much better shape. I think the offline RL and NeurIPS community would benefit from the findings in this paper and deserves to be accepted.

I would still recommend the authors to clarify the points that authors were confused in the paper as they did in the rebuttal and include some of the results presented in the rebuttal in the main paper.




**Award:**

No

---

### Decision · Program_Chairs · 2022-09-14

Accept